# MEANINGFULLY DEBUGGING MODEL MISTAKES USING CONCEPTUAL COUNTERFACTUAL EXPLANATIONS

## ABSTRACT

Understanding and explaining the mistakes made by trained models is critical to many machine learning objectives, such as improving robustness, addressing concept drift, and mitigating biases. However, this is often an ad hoc process that involves manually looking at the model's mistakes on many test samples and guessing at the underlying reasons for those incorrect predictions. In this paper, we propose a systematic approach, *conceptual counterfactual explanations* (CCE), that explains why a classifier makes a mistake on a particular test sample(s) in terms of human-understandable concepts (e.g. this zebra is misclassified as a dog because of faint *stripes*). We base CCE on two prior ideas: counterfactual explanations and concept activation vectors, and validate our approach on well-known pretrained models, showing that it explains the models' mistakes meaningfully. In addition, for new models trained on data with spurious correlations, CCE accurately identifies the spurious correlation as the cause of model mistakes from a single misclassified test sample. On two challenging medical applications, CCE generated useful insights, confirmed by clinicians, into biases and mistakes the model makes in real-world settings. The code for CCE is publicly available and can easily be applied to explain mistakes in new models.

## 1 INTRODUCTION

People who use machine learning (ML) models often need to understand why a trained model is making a particular mistake. For example, upon seeing a model misclassify an image, an ML practitioner may ask questions such as: *Was this kind of image underrepresented in my training distribution? Am I preprocessing the image correctly? Has my model learned a spurious correlation or bias that is hindering generalization?* Answering this question correctly affects the *usability* of a model and can help make the model more *robust*. Consider a motivating example:

**Example 1: Usability of a Pretrained Model**. A pathologist downloads a pretrained model to classify histopathology images. Despite a high reported accuracy, he finds the model performing poorly on his images. He investigates why that is the case, finding that the **hues** in his images are different than in the original training data. Realizing the issue, he is able to transform his own images with some preprocessing, matching the training distribution and improving the model's performance.

In the above example, we have a domain shift occurring between training and test time, which degrades the model's performance (Subbaswamy et al., 2019; Koh et al., 2020). By *explaining* the cause of the domain shift, we are able to easily fix the model's predictions. On the other hand, if the domain shift is due to more complex spurious correlations the model has learned, it might need to be completely *retrained* before it can be used. During development, identifying and explaining a model's failure points can also make it more robust (Abid et al., 2019; Kiela et al., 2021), as in the following example:

**Example 2: Discovering Biases During Development**. A dermatologist trains a machine learning classifier to classify skin diseases from skin images collected from patients at her hospital. She shares her trained model with a colleague at a different hospital, who reports that the model makes many mistakes with his own patients. She investigates why the model is making mistakes, realizing that patients at her colleague's hospital have different **skin colors** and **ages**. Answering this question allows her to not only know her own model's biases but also guides her to expand her training set with the right kind of data to build a more robust model.

Explaining a model's mistakes is also useful in other settings where data distributions may change, such as concept drift for deployed machine learning models (Lu et al., 2018). Despite its usefulness, explaining a model's performance drop is often an ad hoc process that involves manually looking at the model's mistakes on many test samples and guessing at the underlying reasons for those incorrect predictions. We present *counterfactual conceptual explanations* (CCE), a systematic method for explaining model mistakes in terms of meaningful human-understandable concepts, such as those in the examples above (e.g. *hues*). In addition, CCE was developed with the following criteria:

- **No training data or retraining**: CCE does not require access to the training data or model retraining.
- **Only needs the model**: CCE only needs white-box access to the model. The user can use any dataset to learn desired concepts.
- **High-level explanations**: CCE provides high-level explanations of model mistakes using concepts that are easy for users to understand.

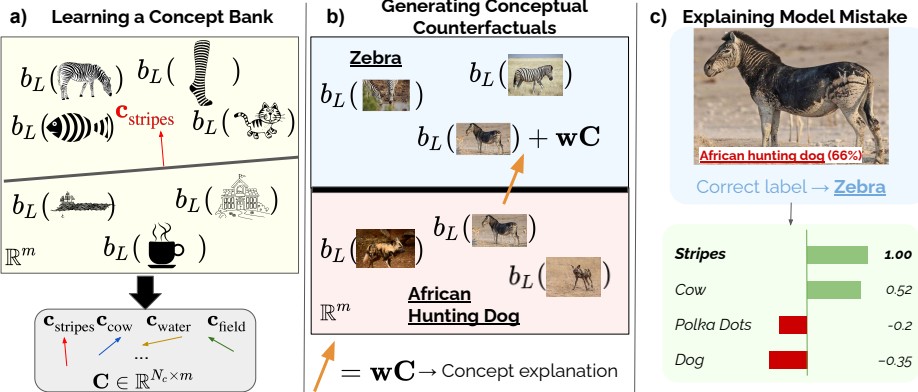

Figure 1: **Explaining with Conceptual Counterfactuals: (a)** In generating conceptual explanations, the first step is to define a concept library. After defining a concept library, we then learn a concept activation vector (CAV) (Kim et al., 2018) for each concept. **(b)** Given a misclassified sample, such as the Zebra image shown here as an African hunting dog, we would like to generate a conceptual counterfactual. Meaning that we would like to generate a perturbation in the embedding space that would correct the model prediction, using a weighted sum of our concept bank. **(c)** Our method assigns a score to the set of concepts. A large positive score means that adding that concept to the image (e.g. *stripes*) will increase the probability of correctly classifying the image, as will removing or reducing a concept with a large negative score (e.g. *polka dots* or *dogness*).

Fig. 1 provides an overview of our CCE method. To summarize **our contributions**, we develop a novel method, CCE, that can systematically explain model mistakes in terms of high-level concepts that are easy for users to understand. We show that CCE correctly identifies spurious correlations in model training, and we quantitatively validate CCE across different experiments over natural and clinical images. We have released all of the code and data needed for our method in a public repository: `https://github.com/conceptualcounterfactuals/iclr2022`.

## 2 RELATED WORKS

CCE is inspired by prior efforts to explain machine learning models' predictions. Two ideas are particularly relevant: *counterfactual explanations* and *concept activation vectors*. Here, we provide an overview of these methods and discuss the differences between them and our proposed approach.

**Counterfactual Explanations** Counterfactual explanations (Verma et al. (2020) provides a comprehensive review) are a class of model interpretation methods that seek to answer: *what perturbations to the input are needed for a model's prediction to change in a particular way?* A large number of counterfactual explanation methods have been proposed with various desiderata such as sparse perturbations (Wachter et al., 2017), perturbations that remain close to the data manifold (Dhurandhar

et al., 2018), and causality (Mahajan et al., 2019). What these methods share in common is that they provide an interpretable link between model predictions and perturbations to input features. There are several works focusing on counterfactual explanations for image data(Goyal et al., 2019; Singla et al., 2020) and many of these methods use distractor images or modify input images to explain the model behavior. However, perturbations or saliency maps in the input space are shown to be tricky to interpret and be adopted by users(Alqaraawi et al., 2020; Adebayo et al., 2018).

To compute CCE, we use a similar approach, perturbing input data to change its predictions, but for a complementary goal: to understand the limitations and biases of our model and its training data, rather than to change a specific prediction. Further improving upon existing work, we do this by directly using human-interpretable concepts, without having access to training data.

**Concept Activation Vectors**    To explain an image classification model's mistakes in a useful way, we need to operate not with low-level features, but with more meaningful concepts. Concept activation vectors (CAVs) (Kim et al., 2018) are a powerful method to understand the internals of a network in human-interpretable concepts. CAVs are linear classifiers trained in the bottleneck layers of a network and correspond to concepts that are usually user-defined or automatically discovered from data (Ghorbani et al., 2019). Unlike many prior interpretation methods, explanations produced by CAVs are in terms of high-level, human-understandable concepts rather than individual pixels (Simonyan et al., 2013; Zhou et al., 2016) or training samples (Koh & Liang, 2017).

In previous literature, CAVs have been used to test the association between a concept and the model's prediction on a class of training data (Kim et al., 2018) as well as provide visual explanations for predictions (Zhou et al., 2018). Our approach extends CAVs by showing that *perturbations* along the CAV can be used to change a model's prediction on a specific test sample, e.g. to correct a mistaken prediction, and thereby be used in a similar manner to counterfactuals. Since samples (even of the same class) can be misclassified for different reasons (see Fig. 6), our approach allows a more contextual understanding of a model's behavior and mistakes.

**Bias/Error Detection and Robustness**    Understanding a model's behavior and explaining its mistakes is critical for building more robust and less biased models, as we have discussed in Section 1. While some other methods, such as FairML (Adebayo & Kagal, 2016) and the What-If Tool (Wexler et al., 2020), can also be used to discover biases and failure points in models, they are typically limited to tabular datasets where sensitive features are explicitly designated, unlike our approach which is designed for unstructured data and works with a more flexible and diverse set of concepts. In a similar spirit to our work, Singh et al. (2020) aim to identify and mitigate contextual bias, however, they assume access to the whole training setup and data; which is a limiting factor in practical scenarios.

Our approach can also be used to explain a model's performance degradation with data drift. It is complementary to methods for out-of-distribution sample detection (Hendrycks & Gimpel, 2016) and black-box shift detection (Lipton et al., 2018), which can detect data drift, but cannot explain the underlying reasons. There are also algorithmic approaches to improve model robustness with data drift (Raghunathan et al., 2018; Nestor et al., 2019). Although these methods can provide some guarantees around robustness for unstructured data, they are in terms of low-level perturbations and do not hold with more conceptual and natural changes in data distributions. There are several recent methods for detecting clusters of mistakes made by a model (Kim et al., 2019; d'Eon et al., 2021), which is useful for detecting disparity in the model, for example. CCE differs in that instead of *finding* mistake clusters, CCE *explains* errors from just one (or a few) image.

## 3    METHODS

In this section, we detail the key steps of our method. Let us define basic notations: let $f : \mathbb{R}^d \to \mathbb{R}^k$ be a deep neural network, let $\boldsymbol{x} \in \mathbb{R}^d$ be a test sample belonging to class $y \in \{1, \ldots k\}$. We assume that that the model misclassifies $\boldsymbol{x}$, meaning that $\arg\max_i f(\boldsymbol{x})_i \neq y$ or simply that the model's confidence in class $y$, $f(\boldsymbol{x})_y$, is lower than desired. Let $m$ be the dimensionality of the bottleneck layer $L$, and $b_L : \mathbb{R}^d \to \mathbb{R}^m$ be the "bottom" of the network, which maps samples from the input space to the bottleneck layer, and $t_L : \mathbb{R}^m \to \mathbb{R}^k$ be the "top" of the network, defined analogously. For readability, we will usually underline class names and *italicize* concepts throughout this paper.

## 3.1 Learning Concepts

In generating conceptual explanations, the first step is to define a concept library: a set of human-interpretable concepts $C = \{c_1, c_2, ...\}$ that occur in the dataset. For each concept, we collect positive examples, $P_{c_i}$, that exhibit the concept, as well as negative examples, $N_{c_i}$, in which the concept is absent. Unless otherwise stated, we use $P_{c_i} = N_{c_i} = 100$. These concepts can be defined by the researcher, by non-ML domain experts, or even learned automatically from the data (Ghorbani et al., 2019). Since concepts are broadly reusable within a data domain, they can also be shared among researchers. For our experiments with natural images, we defined 170 general concepts that include (a) the presence of specific objects (e.g. *mirror*, *person*), (b) settings (e.g. *street*, *snow*) (c) textures (e.g. *stripes*, *metal*), and (d) image qualities (e.g. *blurriness*, *greenness*). Many of our concepts were adapted from the BRODEN dataset of visual concepts (Fong & Vedaldi, 2018). Note that the data used to learn concepts can differ from the data used to train the ML model that we evaluate.

After defining a concept library, we then learn an SVM and the corresponding CAV for each concept. We follow the same procedure as in (Kim et al., 2018), for the penultimate layer of a ResNet18 pretrained in ImageNet. This step only needs to be done once for each model that we want to evaluate, and the CAVs can then be used to explain any number of misclassified samples. We refer the reader to Appendix A.1 for implementational details and Appendix B for a full list of concepts. We denote the vector normal to the classification hyperplane boundary as $c_i$(normalized, i.e. $|c_i| = 1$) and the intercept of the SVM as $\phi_i$. To measure whether concepts are successfully learned, we keep a hold-out validation set and measure the validation accuracy, disregarding concepts with accuracies below a threshold (0.7 in our experiments, which left us with 168 of the 170 concepts). We provide more details about the threshold in Appendix B.

## 3.2 Conceptual Counterfactual Explanations

Drawing inspiration from the counterfactual explanations literature (Verma et al., 2020), we generate a perturbation for a given misclassified test sample by varying the amount of different concepts in a way such that the perturbation satisfies the following principles:

1. **Correctness**: A counterfactual is considered correct if it achieves a desired outcome. In our case, this would mean that the perturbed test point should be classified as the correct label.
2. **Validity**: We would like our counterfactuals to be valid, such that they would not violate real-world conditions. In our case, this means ensuring that the perturbed points contain realistic levels of each concept, as discussed below.
3. **Sparsity**: The ultimate goal of generating explanations is communicating them to users. Analyzing a large number of modifications and interactions may not be trivial, so perturbations should change a small number of concepts.

Let $\mathcal{L}_{CE}$ denote the cross entropy loss. Using the concept vectors and intercepts $(c_i, \phi_i)$(see 3.1), we build our concept bank $C \in \mathbb{R}^{N_c \times m}, \phi \in \mathbb{R}^{N_c}$ where $N_c$ is the number of concepts and $m$ is the output dimension of the bottleneck layer. As our goal is to come up with a scoring scheme for concepts, we need to define what it means to add a concept. For this purpose, we use statistics of training samples. We compute the geometric margin to the decision boundary of the SVM, $d_i = c_i b_L(x_i)^T + \phi_i$, for all of the training examples. As different concepts have different embedding volumes, we scale each concept by the maximum amount of the concept observed in the data used to learn that concept. Let $d_i^{\max}$ denote the maximum margin in the training distribution. We let $\tilde{c}_i = d_i^{\max} c_i$, and using the scaled concept vectors we construct our final concept bank $\tilde{C}$. For example, adding 1 unit of $\tilde{c}_{\text{redness}}$ would mean adding the maximum amount of *redness* seen before.

Our optimization problem is implemented as follows:

$$\min_{\boldsymbol{w}} \quad \mathcal{L}_{\text{CE}}(y, t_L(\mathbf{b}_L(\boldsymbol{x}) + \boldsymbol{w}\tilde{\boldsymbol{C}})) + \alpha|\boldsymbol{w}|_1 + \beta|\boldsymbol{w}|_2$$
$$\text{s.t.} \quad \boldsymbol{w}^{\min} \leq \boldsymbol{w} \leq \boldsymbol{w}^{\max} \tag{1}$$

By minimizing the cross entropy loss, we aim to flip the label of the model prediction to correct the misclassification, to ultimately achieve **correctness**. We do this by adding a weighted sum of concept vectors, weighted by the parameter $\boldsymbol{w}$. Additionally, we apply elastic net regularization to introduce **sparsity** in the concept scores.

We further introduce **validity** constraints to make sure that the concept additions are within a realistic range. Concretely, assume that we have a concept that already exists in the image. We can query this fact by looking at the prediction of our concept SVM. If the concept already exists in the image, adding that concept to the image would be less meaningful. Similarly, if a concept is already absent in the image, then removing it should not be a valid action. Following these intuitions, we will use the bounds $[\mathbf{w}^{\min}, \mathbf{w}^{\max}]$ to guide the optimization.

First, we should not be able to add the concept to explain the model's mistake if the concept is already in the image, i.e. the score $w_i$ should not be positive:

$$w_i^{\max} = 0 \quad \text{if} \quad \boldsymbol{c}_i b_L(\boldsymbol{x})^T + \phi_i > \kappa_i \tag{2}$$

Here, $\kappa_i$ is an offset we use to predict the existence of the concept. Setting $\kappa_i = 0$ would mean using the SVM as the decision boundary. As an alternative strategy, we can vary $\kappa_i$ to allow for weaker forms of validity regularization (e.g. by setting it to the mean positive margin over the training samples).

Moreover, we should be able to restrict the amount of concept we are adding to the embedding, e.g. adding an infinite amount of *redness* may not be meaningful. We first use the training samples to identify the maximum geometric margin for the concept i, $d_i^{\max}$. Then we restrict $w_i$ such that the concept addition would result in a margin at most as large as the maximum margin over the training samples. We want to have

$$\boldsymbol{c}_i^T \left( b_L(\boldsymbol{x}) + w_i \tilde{\boldsymbol{c}}_i \right) + \phi_i \leq d_i^{\max} \tag{3}$$

and thus

$$w_i \leq \frac{d_i^{\max} - \boldsymbol{c}_i^T b_L(\boldsymbol{x}) - \phi_i}{\boldsymbol{c}_i^T \tilde{\boldsymbol{c}}_i} = \frac{d_i^{\max} - \boldsymbol{c}_i^T b_L(\boldsymbol{x}) - \phi_i}{d_i^{\max}} \tag{4}$$

Combining this with the Equation 2 would lead to our final constraint:

$$w_i^{\max} = \begin{cases} 0 & \text{if} \quad \boldsymbol{c}_i b_L(\boldsymbol{x})^T + \phi_i > \kappa_i \\ \frac{d_i^{\max} - \boldsymbol{c}_i^T b_L(\boldsymbol{x}) - \phi_i}{d_i^{\max}} & \text{else} \end{cases} \tag{5}$$

In a similar fashion, we calculate the lower bounds on scores as:

$$w_i^{\min} = \begin{cases} 0 & \text{if} \quad \boldsymbol{c}_i^T b_L(\boldsymbol{x})^T + \phi_i < -\kappa_i \\ \frac{d_i^{\min} - \boldsymbol{c}_i^T b_L(\boldsymbol{x}) - \phi_i}{d_i^{\min}} & \text{else} \end{cases} \tag{6}$$

We solve the problem in Equation 1 using Projected Gradient Descent, where we introduce projection steps to enforce the **validity** constraints. Namely, after each gradient step, we clamp the values of the scores to remain within the precomputed range. The final Conceptual Counterfactual Explanations (CCE) algorithm is given in Algorithm 1. Throughout the experiments, unless otherwise stated, we use $\alpha = 0.1, \beta = 0.01, \gamma = 0.01, \eta = 0.9, \kappa_i = 0$.

---

**Algorithm 1:** Conceptual Counterfactual Explanations(CCE)

---

**Input:** $\boldsymbol{x}, y, [\boldsymbol{w}^{\min}, \boldsymbol{w}^{\max}], \tilde{\boldsymbol{C}}$
**Hyperparameters:** $\alpha, \beta, \gamma, \eta$
**Output:** $\boldsymbol{w}$

1 **while** *w not converged* **do**
2      $\hat{y} \leftarrow t_L(b_L(\boldsymbol{x}) + \boldsymbol{w}\tilde{\boldsymbol{C}})$
3      $\mathcal{L}_{\text{total}} \leftarrow \mathcal{L}_{\text{CE}}(\hat{y}, y) + \alpha|\boldsymbol{w}|_1 + \beta|\boldsymbol{w}|_2$
4      $\boldsymbol{w} \leftarrow \text{GradientDescent}(\mathcal{L}_{\text{total}}, \boldsymbol{w}, \text{lr} = \gamma, \text{momentum} = \eta)$
5      $\boldsymbol{w} \leftarrow \text{clamp}(\boldsymbol{w}, \boldsymbol{w}^{\min}, \boldsymbol{w}^{\max})$

---

In summary, we propose that using our validity constraints and achieving a correct counterfactual, we can explain the model mistakes and behavior in an interpretable (sparse) manner. A large positive score means that adding that concept to the image will increase the probability of correctly classifying the image, as will removing or reducing a concept with a large negative score. CCE provides us with an assessment of which concepts explain a misclassified sample.

## 4   RESULTS

In this section, we demonstrate how CCE can be used to explain model limitations. First, we show that CCE reveals high-level spurious correlations learned by the model. We then show analogous results for low-level image characteristics. Finally, we show real-world medical applications where we are able to identify biases and spurious correlations in the training dataset and give users feedback about the image quality. In all scenarios, CCE (Alg. 1) correctly identifies biases in the model and artifacts in the image as explanations of model mistakes.

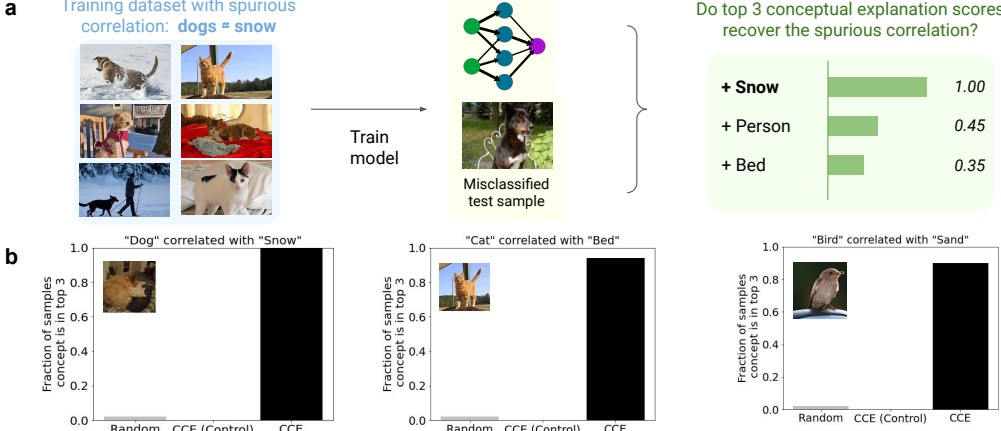

Figure 2: **Validating CCE by Identifying Spurious Correlations: (a)** First, we train 5-class animal classification models on skewed datasets that contain spurious correlations that occur in practice. For example, one model is trained on images of dogs that are all taken with *snow*. This causes the model to associate snow with dogs and incorrectly predicts dogs without snow to not be dogs. We test whether our method can discover this spurious correlation. **(b)** We repeat this experiment with 20 different models, each that has learned a different spurious correlation, finding that in most cases the model identifies the spurious correlation in more than 90% of the misclassified test samples. For comparison, we also run CCE using a control model without the spurious correlation, and we randomly select concepts as well. The random performance is evaluated by sampling three concepts out of the available 150 and calculating the Precision@3.

### 4.1   CCE REVEALS SPURIOUS CORRELATIONS LEARNED BY THE MODEL

We start by demonstrating that CCE correctly identifies high-level spurious correlations that models may have learned. For example, consider a training dataset that consists of images of different animals in natural settings. A model trained on such a dataset may capture not only the desired correlations related to the class of animals but spurious correlations related to other objects present in the images and the setting of the images. We use CCE to identify these spurious correlations.

To systematically validate CCE, we need to know the ground-truth spurious correlations that a model has learned. To this end, we train models with intentional and known spurious correlations using the MetaDataset (Liang & Zou, 2021), a collection of labeled datasets of animals in different settings and with different objects. We construct 20 different training scenarios, each consisting of 5 animal classes (cat, dog, bear, bird, elephant). We trained a separate model for each scenario. In each scenario, one class is only included with a specific confounding variable (e.g. all images of dogs are with *snow*), inducing a spurious correlation in the model (images of dogs without *snow* will be misclassified), which we can probe with CCE. We also train a control model using random samples of animals across contexts, without intentional spurious correlations. In Appendix A.7.2, we provide training distributions with less severe spurious correlations, i.e. where instead of all images having the confounding concept, we use a varying level of severity in the correlations.

Our experimental setup is shown in Fig. 2(a). We train models with $n = 750$ images, fine-tuning a pretrained ResNet18 model. In all cases, we achieve a validation accuracy of at least 0.7. We then present the models with 50 out-of-distribution (OOD) images (randomly sampled from the entire MetaDataset), i.e. images of the class without the confounding variable present during training.

We then use CCE to recover the top 3 concepts that would explain a model's mistake on each of 50 OOD images that are misclassified, then we report if the spurious concept is among these 3 concepts(Precision@3). For almost all images, CCE identifies the ground-truth spurious correlation as one of the top 3 concepts (Fig. 2(b)). For example, in the model we trained with dogs confounded with *snow*, CCE recovered the spurious correlation in 100% of test samples. We compare these results to performing CCE using the control model, in which the same test images are presented, as well as to picking concepts randomly, both of which result in the spurious correlation being identified much less frequently. We provide results in Appendix A.7.3, where instead of doing a sample-by-sample analysis we propose Batch-Mode CCE to analyze a set of mistakes to provide a holistic understanding of the biases in the model.

Additionally, we compare our approach to a (simpler) univariate version of CCE, CCE(Univariate). Instead of running our optimization method over multiple concepts simultaneously, we iterate over each CAV and quantify how a perturbation in the direction of the concept ($d_i^{\max} c_i$) changes the prediction probability of the correct class $y$. Specifically, we compute the CCE as the difference $t_L(b_L(x) + d_i^{\max} c_i)_y - t_L(x)_y$, for each concept, then we order the concepts by the change in the probability.

| Method | Mean Prec@3 | Median Rank |
|---|---|---|
| Random | 0.02 | $82.65(42.7, 120.4)$ |
| CCE(Control) | 0.04 | $32.3(28.03, 40.05)$ |
| CCE(Univariate) | 0.91 | $2.00(1.71, 2.35)$ |
| CCE | 0.95 | $1.85(1.80, 2.10)$ |

Table 1: Empirical evaluation for detecting spurious correlations in training data. We report Precision@3 and ranks averaged over 20 scenarios, the complete table is in Appendix Table 2. Distribution of the Precision@K metric as we vary K can be found in Appendix A.8.

Table 1 provides results over the 20 scenarios and the detailed results can be found in Appendix Table 2. We further provide the mean of the median rank of the concept in each scenario, along with the mean of the first and third quartiles of ranks. Both CCE(univariate) and CCE recover the ground-truth spurious correlation as one of the top 3 concepts correctly across the scenarios. The complete (multivariate) CCE achieves the highest precision and the best median rank. In Appendix A.7.1 we provide results in more challenging scenarios, where the target spurious concept does not exist in the concept bank.

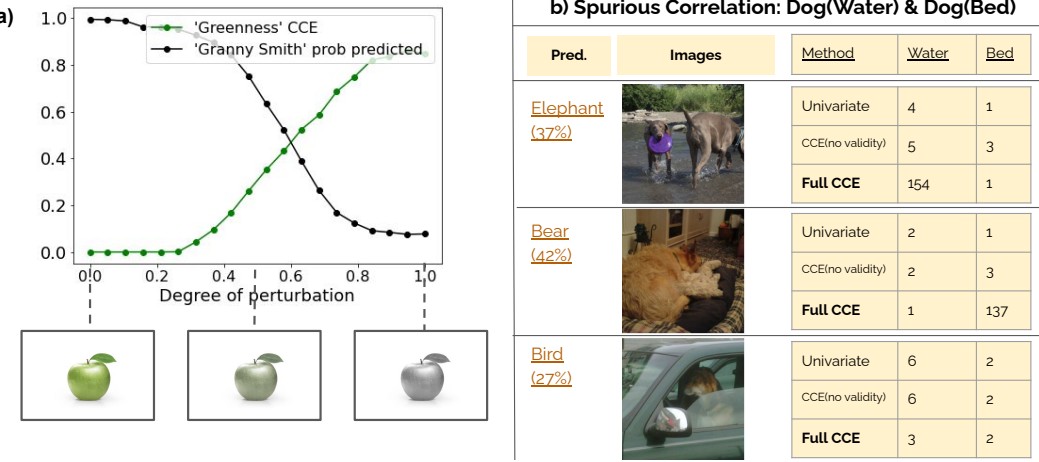

Figure 3: **Validating CCE: (a)** Here, we take an arbitrary test sample of a Granny Smith apple that is originally correctly classified and perturb the image by turning it gray. We compute CCE scores at each perturbed image and observe that the score for *greenness* increases as the image is grayed. **(b)** We demonstrate the effectiveness of validity constraints in a qualitative scenario. In the last two columns, we provide ranks of each concept when a particular method is used. Without validity, methods can use concepts that already exist in the image to explain model mistakes.

## 4.2 CCE REVEALS LOW-LEVEL IMAGE ARTIFACTS

We show that CCE can capture low-level spurious correlations that models may have learned. For example, the ImageNet(Deng et al., 2009) dataset on which SqueezeNet was trained includes a class of green apples known as Granny Smith, which were always colored images. Fig. 3(a) shows that as

the image is grayed, the probability of it being classified as Granny Smith decreases, while the CCE for *greenness* increases. We provide details and examples about this experiment in Appendix A.6.

### 4.3 VALIDITY CONSTRAINT IMPROVES EXPLANATIONS

Here we demonstrate **validity** constraints are necessary to plausibly explain model mistakes. In these experiments, instead of training the model with a single concept that is associated with a class, we use two concepts. For instance, we use 50 dog images with *water* and 50 with *bed* during training. In Figure 3a, we show such a scenario. In all of these images, CCE(Univariate) identifies both *bed* and *water* concepts to explain the model mistakes. However, in the first image, there is already *water* and in the second image, there is already a *bed*. Thus, using them in the counterfactual result in an invalid explanation. However, when CCE with **validity** constraints is used, the score for the *water* concept in the first image and the *bed* concept in the second image drops. Ultimately, this would result in a plausible counterfactual explanation. More examples are provided in Appendix A.3.

### 4.4 EVALUATING CCE IN THE WILD

We run experiments where we evaluate CCE on real-world medical data and models. We demonstrate that CCE helps identify learned biases and artifacts that provide insight into a model's mistakes. Throughout our evaluations, we worked with a board-certified dermatologist and cardiologist to confirm the clinical relevance of CCE explanations.

**Dermatology - Skin Condition Classification:** We follow the model training procedure described in (Groh et al., 2021) to train a ResNet18 model to predict one of the 114 skin conditions using the Fitzpatrick17k dataset of 16,577 annotated skin images (Groh et al., 2021). This classification model achieved 20% overall accuracy, closely matching the number reported in the paper (see Appendix A.4 for details). To explain the model's mistakes, in addition to the 168 concepts used in Sec. 3.1, we also learned 8 clinically relevant concepts: *defocus blur*, *zoom blur*, *brightness*, *motion blur*, *contrast*, *dark skin type*, *skin hair*, and *zoom*. To learn each of these concepts, we use 25 pairs of positive and negative images (except the *skin hair* concept, for which we used 10 images, where the skin hair images are obtained from the ISIC (Rotemberg et al., 2021) dataset). In Figure 4, we observe several

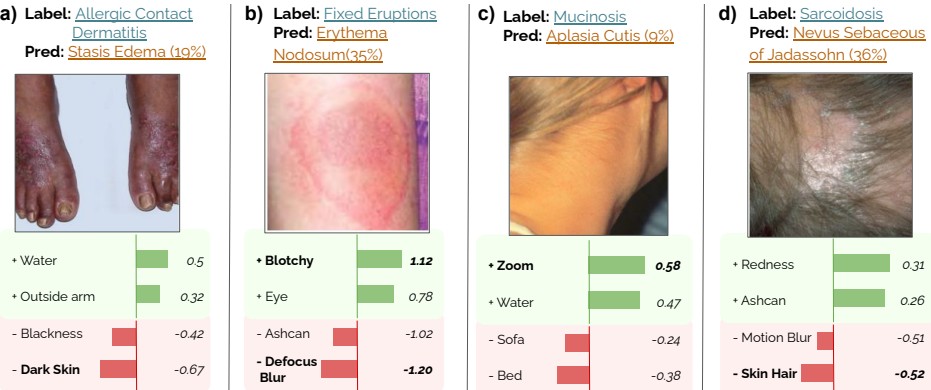

Figure 4: **CCE explains model mistakes using learned biases and image quality conditions. (a)** CCE identifies *dark skin type* correlation with the allergic contact dermatitis condition that exists in the training dataset. **(b, c, d)** CCE identifies image artifacts that degrade the model performance.

ways CCE guides our understanding of model mistakes. In addition to the unbalanced fraction of skin type groups over the whole dataset, we find that there are wider discrepancies when particular skin conditions are considered. For example, for the allergic contact dermatitis condition, there are 259 images from the lightest skin tones in the training data, 137 images of intermediate skin tones, and only 24 images of dark skin tones. In the test set, for 23 out of all 24 allergic contact dermatitis dark-skin images where the model makes a mistake, CCE identifies the *dark skin type* concept as one of the 3 concepts with the largest negative score. This collectively reflects bias in the training data.

For fixed eruptions and mucinosis, CCE finds image qualities that would increase the classification performance. Namely, CCE identified that the blur in 2nd image in Fig. 4 caused the model's mistake

(reducing *blur* would correct the mistake). In the 3rd image, CCE learned that the image is too zoomed out, and increasing *zoom* would correct the mistake. For the 4th image, CCE identified that too much *skin hair* in the image contributed to the model's mistake. This is consistent with other studies which show that presence of skin hair can degrade the model performance (Okur & Turkan, 2018). In all cases, CCE identifies relevant concepts from our expanded concept bank and displays them to the user. These explanations were validated by a trained dermatologist as plausible reasons why these images may have been difficult to classify. In Appendix A.4, we describe details of this experiment and include comments about concepts suggested by CCE that seem irrelevant.

**Cardiology - Pneumothorax Classification from Chest X-Ray Images**  Several studies have raised concerns about the biases learned by chest X-ray diagnosis models and their deployability in novel domains (Seyyed-Kalantari et al., 2020; Larrazabal et al., 2020; Wu et al., 2021). We investigate a cross-site evaluation setting, where models trained on Chest X-Ray images are used to classify the pneumothorax condition. Specifically, we are taking a binary classification model trained on a dataset collected from the National Institutes of Health Clinical Center in Bethesda (NIH)(Wang et al., 2017) and we test the model with images obtained from the Stanford Health Care in Palo Alto (SHC)(Irvin et al., 2019). In this experiment, we follow the protocol described in (Wu et al., 2021); see Section A.5 for details. Notably, the NIH dataset consists of images from the frontal (AP/PA) view positions, whereas in the SHC dataset, there are also images obtained from the Lateral View. This would mean that the model has not seen any images from the lateral view during training. Several examples of these images can be seen in Appendix Figure 10. Similar to Sec.4.4, we use 100 pairs of chest X-ray images to learn clinically relevant concepts: *Lateral View* & *AP View*(view positions), *Cardiomegaly* & *Atelectasis*(comorbidities), *patient gender*, and *patient age under 24*.

From the SHC dataset, we randomly select 150 images taken from the lateral view where the model makes a mistake. We run CCE using a concept bank with 120 BRODEN concepts, in addition to our clinically relevant concepts. In **100**% of these 150 test images, the *Lateral View* concept received the largest negative score, meaning that CCE finds removing the concept would increase the probability of correctly classifying these images. This is consistent with cardiologist expectation: a model that has not seen images from the lateral view may fail to generalize to that category. With the help of CCE, end-users can identify these shortcomings and help address the biases in the training dataset.

## 5 Conclusion and Future Work

We present a simple and intuitive method, CCE, that generates meaningful insights into why machine learning models make mistakes on test samples. We validated that CCE identifies spurious correlations learned by the model for both natural images and medical images, where CCE's explanations are confirmed by clinicians. In medical applications, CCE was able to inform the end-user about the image quality conditions and identify biases in the model's training data. CCE is fast: the concept bank just needs to be learned once using simple SVMs and each test example takes $< 0.3$ seconds on a single CPU. It can be readily applied to any deep network without retraining and provides explanations of mistakes in human-interpretable terms.

CCE can detect biases in training data that lead to model mistakes without needing the training data. It requires a small number of labeled examples to learn concepts. For instance, the dermatology experiments used 50 images to learn each concept. The data used to learn concepts can come from datasets *different* from the ones used to train the model. In the dermatology case study, we learned concepts using the ISIC dataset (for *skin hair*) and still correctly explained the mistakes of a model trained on the Fitzpatrick17k data. This makes the CCE method more broadly useful. The quality of the concept bank is a crucial component for the outcome of the explanations, as demonstrated in AppendixA.7.1. It is important to seek guidance from the experts of the problems when building concept libraries to obtain the best results. Incorporating automatic concept learning(Ghorbani et al., 2019) is a fruitful future direction, as it could further simplify the entire pipeline.

There are several promising areas where CCE can be extended. While we focused in this paper on image classification tasks, CCE can be applied to other data modalities such as text, audio, and video data, as well as other tasks such as regression and segmentation. Finally, we seek to do user studies to understand how human subjects respond to explanations with CCE and how it drives improvements in model debiasing and robustness.

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

# A    APPENDIX

## A.1    LEARNING CONCEPTS

Here we provide implementational details on how we learn the concept activation vectors (CAVs). Concretely, we pick a bottleneck layer in our model, which is the representation space in which we will learn our features. Unless otherwise stated, we choose the penultimate layer in ResNet18 for experiments in this paper. Let $m$ be the dimensionality of the bottleneck layer $L$, and $b_L : \mathbb{R}^d \to \mathbb{R}^m$ be the "bottom" of the network, which maps samples from the input space to the bottleneck layer, and $t_L : \mathbb{R}^m \to \mathbb{R}^k$ be the "top" of the network, defined analogously. In generating conceptual explanations, the first step is to define a concept library: a set of human-interpretable concepts $C = \{c_1, c_2, ...\}$ that occur in the dataset. For each concept, we collect positive examples, $P_{c_i}$, that exhibit the concept, as well as negative examples, $N_{c_i}$, in which the concept is absent. Then, we train a support vector machine to classify $\{b_L(\boldsymbol{x}) : \boldsymbol{x} \in P_{c_i}\}$ from $\{b_L(\boldsymbol{x}) : \boldsymbol{x} \in N_{c_i}\}$, the same way as in Kim et al. (2018). Overall pseudocode for the procedure can be found in Figure 5.

---

**Algorithm 2:** Learning concept vectors

**Inputs :**
```
      f  # trained network:  model
      L  # bottleneck layer (hyperparameter):  int
      concepts # set of concepts:  set[str]
      P  # positive examples per concept:  dict[str,
      list[sample]]
      N  # negative examples per concept:  dict[str,
      list[sample]]
```
**Return :** svms #Set of SVMs containing concept predictors.

```
1 b, t = f.layers[:L], f.layers[L:] # Divide network f(·) into a
  bottom b (first l layers) and top t (remaining layers) so that
  f(·) = t(b(·))
2 for c in concepts: # Per concept, learn an SVM to classify
  bottleneck representations of positive and negative examples.
3   svms[c] = svm.train(b(P[c]), b(N[c]))
4   # Filter out concepts that are not learned well (i.e.
  validation accuracies below a particular threshold).
5   if svms[c].acc < .7:
6     del svms[c]
```

---

Figure 5: **Pseudocode for CES** in Python-like syntax concept learning procedure. Learning the concepts (lines 1-6) just needs to be done once and can be carried out offline.

## A.2    METADATASET EXPERIMENTS

In Table 2, we provide results over 20 scenarios. In each of the scenarios, we fine-tune only the classification layer of a ResNet18(He et al., 2016) pretrained on ImageNet, which outputs a probability distribution over 5 animals(cat, dog, bear, bird, elephant). For instance, in the case of the dog(snow) experiment, we train the model with images of dogs that are all taken with snow. We replicate this experiment with 20 different animal & concept combinations and report the results below.

In Table 3, for each scenario, we sample 50 images with and without the concept and we report the accuracy over those images. We observe a start difference between accuracies, which verifies that model learns to rely on the correlation.

## A.3    VALIDITY EXAMPLES

In Fig. 7, we provide additional examples on where validity improves explanations. In **a)** we have a model trained with the dog-*snow* spurious correlation and in **b)** we have the dog class associated with both the *horse* and *bed* concepts. For both of these examples, we observe that in the images where

| Experiment | CCE-Prec.@3 | CCE(Univariate)-Prec.@3 | CCE(Control)-Prec.@3 |
|---|---|---|---|
| dog(chair) | 0.980 | 0.360 | 0 |
| cat(cabinet) | 1 | 1 | 0.180 |
| dog(snow) | 1 | 1 | 0 |
| dog(car) | 1 | 1 | 0 |
| dog(horse) | 1 | 1 | 0.500 |
| bird(water) | 0.960 | 1 | 0.660 |
| dog(water) | 0.980 | 0.980 | 0 |
| dog(fence) | 1 | 0.980 | 0 |
| elephant(building) | 1 | 0.821 | 0 |
| cat(keyboard) | 0.860 | 0.800 | 0 |
| dog(sand) | 0.760 | 0.900 | 0 |
| cat(computer) | 0.980 | 1 | 0 |
| dog(bed) | 1 | 1 | 0 |
| cat(bed) | 0.980 | 1 | 0 |
| cat(book) | 0.960 | 1 | 0 |
| dog(grass) | 0.700 | 0.780 | 0 |
| cat(mirror) | 0.900 | 0.900 | 0 |
| bird(sand) | 0.960 | 1 | 0 |
| bear(chair) | 0.940 | 0.720 | 0 |
| cat(grass) | 0.940 | 0.900 | 0 |

Table 2: Empirical evaluation for detecting spurious correlations in the training data. We report results over 20 scenarios, where each class is associated with the concept in parenthesis during the training phase.

| Experiment | Accuracy for Images with the Concept | Accuracy for Images Without the Concept |
|---|---|---|
| dog(chair) | 0.74 | 0.54 |
| cat(cabinet) | 0.8 | 0.62 |
| dog(snow) | 0.76 | 0.12 |
| dog(car) | 0.86 | 0.66 |
| dog(horse) | 0.70 | 0.36 |
| bird(water) | 0.78 | 0.42 |
| dog(water) | 0.82 | 0.4 |
| dog(fence) | 0.74 | 0.52 |
| elephant(building) | 0.82 | 0.72 |
| cat(keyboard) | 0.96 | 0.46 |
| dog(sand) | 0.66 | 0.42 |
| cat(computer) | 1.0 | 0.68 |
| dog(bed) | 0.86 | 0.46 |
| cat(bed) | 0.84 | 0.54 |
| cat(book) | 0.9 | 0.64 |
| dog(grass) | 0.96 | 0.56 |
| cat(mirror) | 0.86 | 0.4 |
| bird(sand) | 0.78 | 0.66 |
| bear(chair) | 0.82 | 0.44 |
| cat(grass) | 0.72 | 0.46 |

Table 3: Accuracy of the model for images tested with and without the confounding variable.

the proposed concept already exists, validity constraints help prevent the model from explaining the mistake by adding that particular concept, which results in valid explanations.

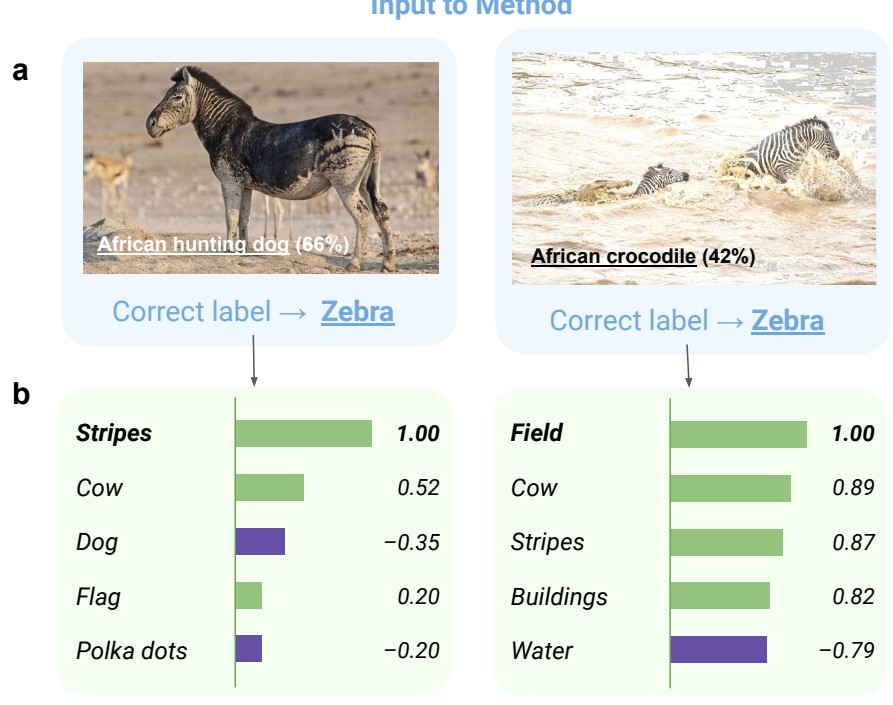

Figure 6: **Different reasons of model mistakes for the same class.**.

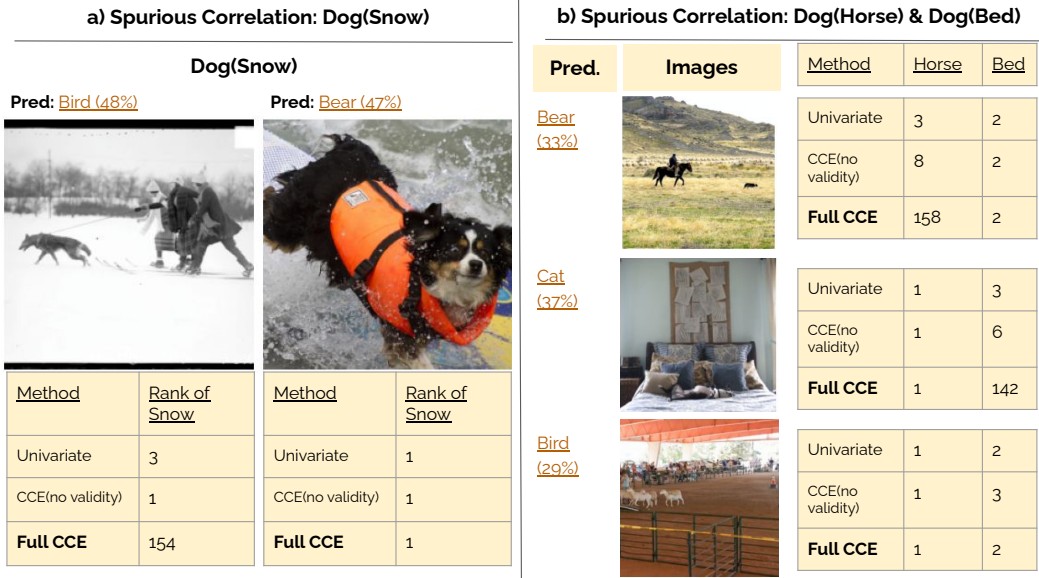

Figure 7: **Validity constraint improves explanations.** .

## A.4 DERMATOLOGY EXPERIMENT WITH FITZPATRICK17K

We directly follow the experimental protocol in Groh et al. (2021). Fitzpatrick17k (Groh et al., 2021) is a dermatology dataset that contains 16,577 skin images with 114 different skin conditions. For

these images, skin type labels are provided using the Fitzpatrick system (Fitzpatrick, 1988). Groh et al. (2021) decomposes the dataset into three groups: The lightest skin types (1 & 2) with 7,755 images, the middle skin types (3 & 4) with 6,089 images, the darkest skin types (5 & 6) with 2,168 images. We randomly partition the dataset into training(80%) and testing(20%) sets. We use a ResNet18 (He et al., 2016) backbone pretrained on ImageNet(Deng et al., 2009) and we fine-tune a classification head to predict one of the 114 skin conditions using Adam(Kingma & Ba, 2014) optimizer. The classification head consists of 1) a fully-connected layer with 256 hidden units 2) relu activation 3) dropout layer with a 40% probability of masking activations 4) another fully-connected layer with the number of predicted categories. We obtain the training script from the repository[1] for the paper Groh et al. (2021).

We are using the *imagecorruptions*[2](Michaelis et al., 2019) library to obtain positive samples of *defocus blur*(3), *zoom blur*(2), *brightness*(3), *motion blur*(3), and *contrast*(2), where we used the severity levels provided in parenthesis. Examples of positive images for these concepts can be found in Fig. 8.

We learn the *zoom* concept by cropping the height and width of images to a fourth of their sizes. We obtain positive images from the darkest skin types (5&6) and negative images from the lightest skin types (1&2) to learn the *dark skin color* concept. For all dermatology concepts except the *skin hair* concept, we use 25 positive & negative pairs of images. For the skin hair concept, we use 10 pairs of images from the ISIC dataset (Rotemberg et al., 2021).

### A.4.1 EXPLANATION ARTIFACTS

It is important to note that sometimes concepts such as *water* or *ashcan* may be suggested as explanations by CCE. Concepts are represented as vectors in the embedding space, and there may be similarities among them. Hence, *water* or *ashcan* concepts may be very similar to certain textures that are relevant to the dermatology task, where it was not possible to obtain the true concept itself. The practitioner using CCE to explain the mistake can easily discard these artifacts suggested by CCE, where there is also concepts that are more relevant and obtained larger scores. Secondly, as the user keeps using concept-based methods, they can understand what type of concepts is needed, or is relevant to the task. Accordingly, it is possible to update and obtain a more informative bank. Generally, there are various interesting directions that could be pursued in the human-machine interface that would improve the real-life performance of explanation algorithms. We are also interested in further understanding this, and think more about the deployment aspects of CCE in future work.

### A.4.2 TESTING GRADCAM++ IN THE FITZPATRICK SETTING

As we mention in Section 4.4, the allergic contact dermatitis condition has a biased skin color distribution in the training data. Activation-map-based methods such as GradCAM++(Chattopadhay et al., 2018) fail to communicate this to the user. However, CCE can correctly identify this as a reason why the model fails. In Figure 9 we show two examples where CCE is able to identify the bias in the data that drives the model mistake; however, one of the methods that are very commonly used to understand model predictions, GradCAM++, fails to identify this reason.

## A.5 CARDIOLOGY EXPERIMENT WITH CHEST X-RAYS

We investigate a cross-site evaluation setting, where models trained on Chest X-Ray images are used to classify the pneumothorax condition which was proposed in (Wu et al., 2021). Using pretrained models from (Wu et al., 2021), we use a DenseNet-121(Huang et al., 2017) architecture pretrained on ImageNet(Deng et al., 2009) and then fine-tuned on the NIH dataset(Wang et al., 2017). As it is reported in their paper, this model achieves 0.779 mean AUC when tested on the (Irvin et al., 2019) dataset and 0.903 mean AUC when tested on the NIH dataset. We use CCE to explain mistake over a subset of the SHC dataset. Specifically, we run CCE over the images taken from the *lateral view*, which does not exist in the training dataset since NIH only contains images from the frontal (Anterior-Posterior(AP) or Posterior-Anterior(PA)) view positions. This constitutes a real-life scenario of

---

[1]`https://github.com/mattgroh/fitzpatrick17k`
[2]`https://github.com/bethgelab/imagecorruptions`

# Positive Samples for Image Quality Concepts

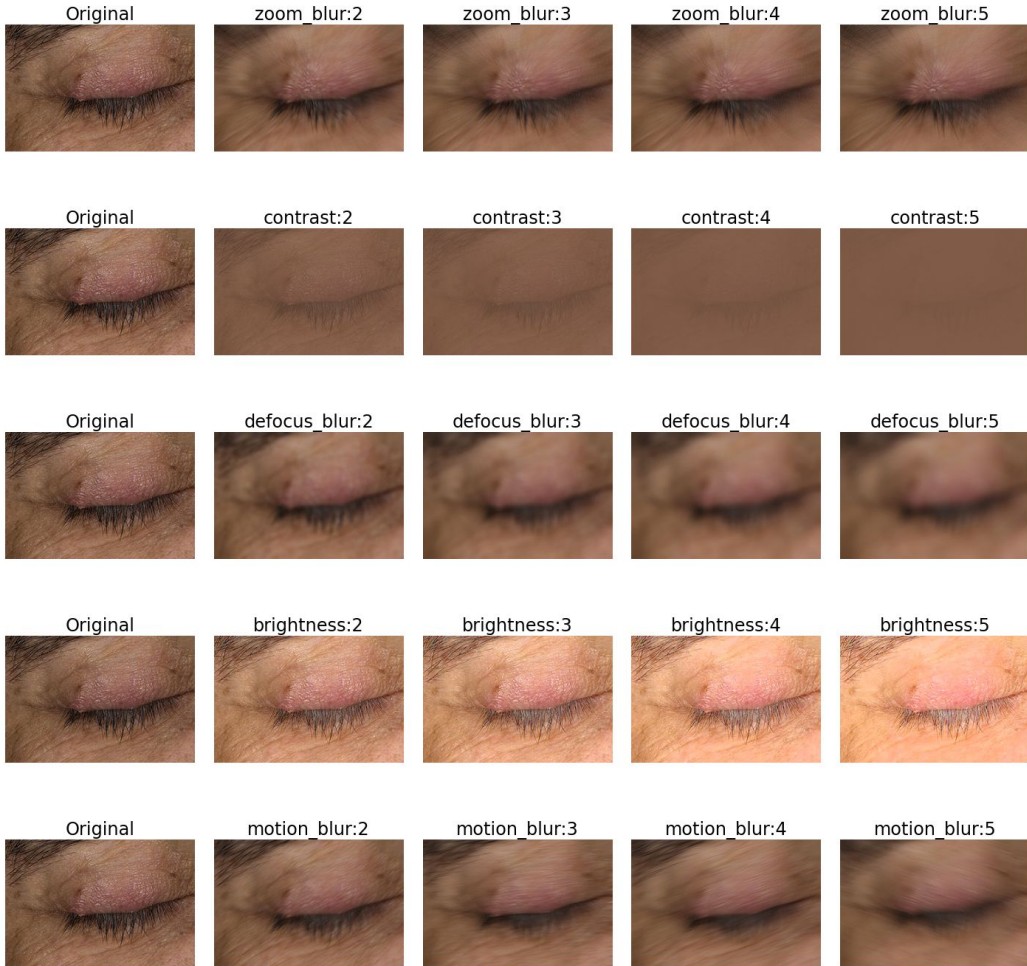

Figure 8: **Learning image quality concepts.** Here we have the positive samples for different image quality concepts. For each corruption type, we provide examples for different levels of severity. The original image is obtained from the Fitzpatrick dataset.

a distribution shift. Some examples of these images can be seen in Figure 10. We sample 150 images that were taken from the *lateral view* where the model makes a mistakes. For **150/150** of those images, CCE identifies the lateral view concept as the concept with the lowest score. Namely, CCE claims that removing the *lateral view*ness concept would increase the probability of correctly classifying the image.

## A.6 ADDITIONAL EXAMPLES VALIDATING CCE THROUGH LOW-LEVEL IMAGE PERTURBATIONS

In Section 4.2, we report a case where CCE reveals low-level artifacts learned by a SqueezeNet(Iandola et al., 2016) pretrained on ImageNet. Particularly, the ImagetNet dataset on which SqueezeNet was trained includes a class of green apples known as Granny Smith. These images were always colored in ImageNet, meaning that the model misclassifies images of these apples in grayscale. We can use this fact to gradually transform a natural image of a Granny Smith apple,

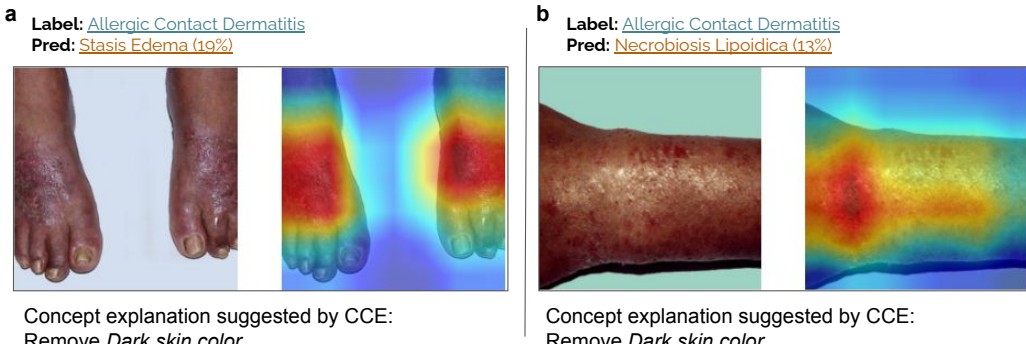

**Figure 9: CAM based methods fail to communicate model biases.** Here we show two examples where CCE is able to identify the bias in the data that drives the model mistake. However, one of the methods that are very commonly used to understand model predictions, GradCam++, fails to identify and communicate the underlying reason.

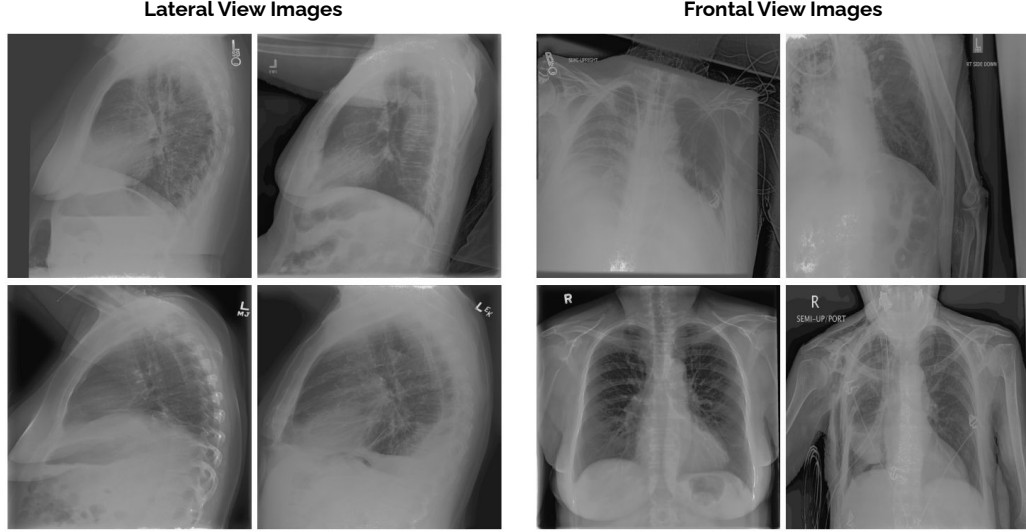

**Figure 10: Chest X-Ray images from different views.** Here we provide several lateral view and frontal view images from the SHC dataset. On the left, we see images from the lateral view and on the right we have images from the frontal view. NIH dataset does not have any lateral view images in the dataset, which can explain why a model trained on the NIH dataset performs poorly when tested on lateral view images.

blending it with its grayscale version. At different levels of the original image vs. its transformed version, we run it through SqueezeNet to obtain a prediction and then calculate its CCE scores. The results, shown in Fig. 3(a), show that as the image is grayed, the probability of it being classified as Granny Smith decreases, while the CCE for *greenness* increases. We repeat this experiment with 25 images of Granny Smith apples and provide results in Appendix Fig. 11(b). Our results show that CCE is also effective at explaining a model's mistakes in terms of low-level visual artifacts.

In Fig. 3, we showed an example in which CCE was successfully able to identify why images that are being perturbed through low-level image transformations are being misclassified. Here, we show another pair of examples, with a different concept: *redness* (Fig. 12). Additionally, in Fig. 11 we show the aggregated curve over 25 granny smith apple images.

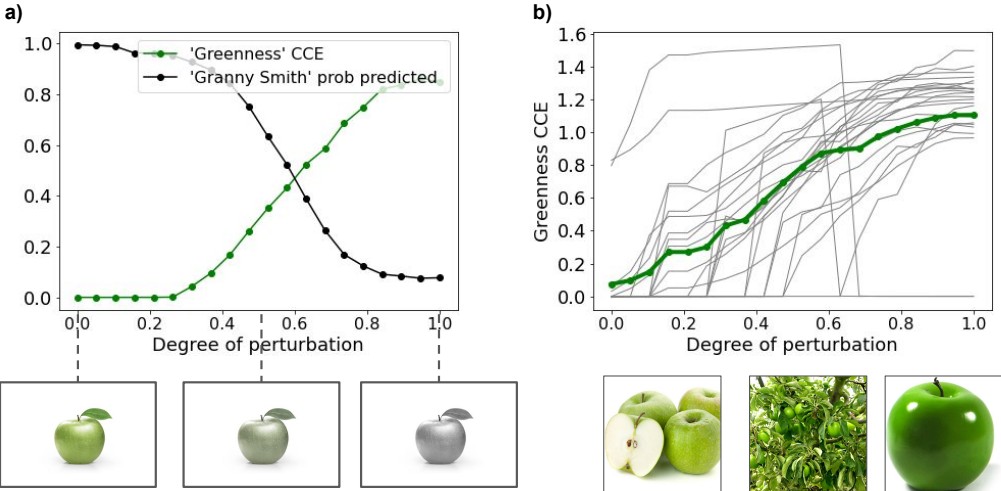

Figure 11: **Validating CCE Through Low-Level Image Perturbations:** **(a)** Here, we take an arbitrary test sample of a Granny Smith apple that is originally correctly classified and perturb the image by turning it gray until it is eventually misclassified (in this case, as mortar). We compute the CCE scores at each perturbed input image and observe that the score for *greenness* increases, corresponding to the degree to which we remove the green color from the image. **(b)** We repeat this for 25 different images of Granny Smith apples (some of which are shown under the plot) and find that the same trends generally hold true (each image is a gray line). Although a few images do not follow this trend, the mean CCE score (bolded green line) does.

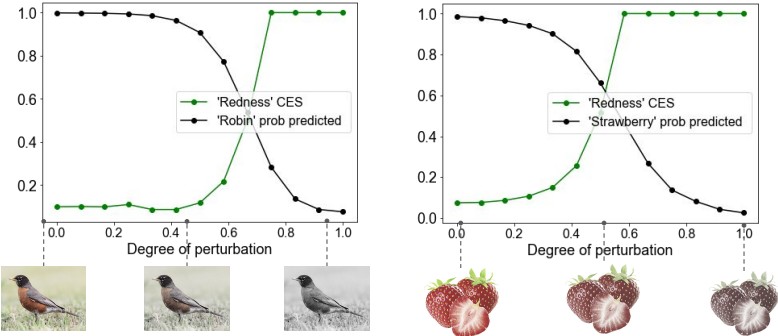

Figure 12: In an analogous manner to Fig. 3, we take images that were originally correctly classified as robin and strawberry, and perturb them by removing red colors until they are misclassified by the model (x-axis). The CCE scores correctly identify the concept of *redness* as the most important concept for correcting the model's mistakes.

## A.7 CHALLENGING SCENARIOS FOR CCE

### A.7.1 WHEN THE TARGET CONCEPT IS MISSING FROM THE BANK

What happens when the spuriously correlated concept is not in our concept bank? In this section, we replicate our controlled experiments in Metadataset by omitting the target concept from our concept bank. For instance, for the dog(snow) experiment, we remove the concept *snow* from the set of 168 concepts that we have and re-run our analysis.

| Experiment | Top5 Concepts |
|---|---|
| dog(bed) | Sofa, Dog, Bedclothes, Muzzle, Headboard |
| dog(chair) | Sofa, Book, Hand, Bedclothes, Dog |
| cat(cabinet) | Inside arm, Microwave, Chest of drawers, Door frame, Oven |
| dog(snow) | Mountain, Car, Dog, Minibike, Headlight |
| dog(car) | Bus, Headlight, Motorbike, Fence, Coach |
| dog(horse) | Cow, Muzzle, Motorbike, Bicycle, Grass |
| bird(water) | Bird, Airplane, Coach, Sand, Mountain |
| dog(water) | Muzzle, Dog, Sand, Airplane, Mountain |
| dog(fence) | Field, Grass, Coach, Muzzle, Dog |
| elephant(building) | Horse, Cow, Pedestal, Car, Bucket |
| cat(keyboard) | Cat, Computer, Inside arm, Paper, Faucet |
| dog(sand) | Water, Airplane, Horse, Blueness, Mountain |
| cat(computer) | Faucet, Keyboard, Cat, Inside arm, Microwave |
| cat(bed) | Cat, Pillow, Inside arm, Headboard, Back pillow |
| cat(book) | Computer, Cat, Bookcase, Oven, Chest of drawers |
| dog(grass) | Horse, Field, Muzzle, Bus, Motorbike |
| cat(mirror) | Inside arm, Headlight, Door frame, Countertop, Bathtub |
| bird(sand) | Water, Airplane, Bird, Mountain, Blind |
| bear(chair) | Food, Candlestick, Plate, Lamp, Fabric |
| cat(grass) | Tree, Mouth, Field, Path, Blind |

Table 4: CCE suggestions when the target concept is missing from the concept bank. We average the ranks for each concept over 50 mistakes in each experiment, and report the Top-5 concepts with the highest rank.

In Table 4, we provide the Top-5 concepts suggested by CCE. We use 50 mistakes for each scenario, evaluate the ranks for each concept in our concept bank using CCE, and average the ranks over all the samples that we have. We report the Top-5 concepts with the highest rank. In almost all of the scenarios, CCE identifies concepts that are 'hinting' to the target concepts. For example, in the dog(bed) case, CCE reports *Sofa, Dog, Bedclothes, Muzzle, Headboard* as the reasons for the mistake, which are either related to the target class or the spuriously correlated concept. Generally, we see a similar pattern in all of our experiments. This provides another piece of evidence that highlights the importance of the richness of the concept bank. If we have a rich-enough bank, then CCE helps us identify the spuriously correlated concept even if it is not directly in the bank.

### A.7.2 WHEN THE CORRELATIONS ARE LESS DRASTIC

What happens when the correlations are less drastic? More concretely, in Section 4.1, we assumed **all** images in the training dataset contains the spuriously correlated concept, we call this situation *100% severity* in this section. Here, we evaluate when the correlation is less severe. Particularly, what happens when we have a scenario with $50\%$ severity, i.e. when only $50\%$ of the training images contain the spuriously correlated concept? In Figure 13a & b, we report the CCE performance as we vary the severity. The bolded line shows the median performance across 20 scenarios, and the confidence levels show the first and third quantiles. We observe that starting from relatively low levels of severity($\approx 20\%$), CCE is able to identify the spuriously correlated concept in a majority of the scenarios. Consequently, we see that CCE can perform well even in less-severe, and thus more realistic scenarios.

### A.7.3 BATCH EVALUATION FOR A SET OF MISTAKES

Here we extend CCE to a batch evaluation setting. Namely, instead of explaining an individual mistake, we aim to explain a batch of mistakes using conceptual counterfactuals. We do so by making a slight modification to our optimization procedure:

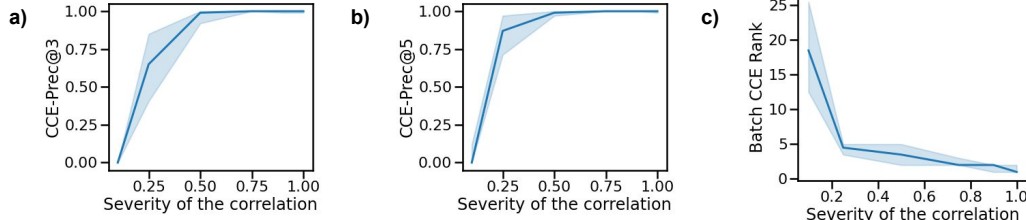

Figure 13: **(a, b)** We report the performance of CCE as we vary the severity level of the correlation.(K% severity means K% of the images in the training dataset has the concept). The bolded line shows the median performance across 20 scenarios, and the confidence levels show the first and third quantiles. We observe that starting from relatively low levels of severity($\approx 20\%$), CCE is able to identify the spuriously correlated concept in a majority of the scenarios. **(c)** We evaluate Batch Mode CCE over controlled experiments with varying severity levels. The y-axis corresponds to the rank of the target concepts, the x-axis gives the severity of the spurious correlation, the bold line is the median performance across 20 scenarios and the confidence intervals give the lower and upper quartiles. We observe that Batch Mode CCE is able to provide an automated analysis of the model biases using a batch of samples, given that it can identify the spuriously correlated concept as one of the major causes over all model mistakes for the given class.

$$\min_{\boldsymbol{w}} \quad \frac{1}{N} \sum_{i=1}^{N} \mathcal{L}_{\text{CE}}(y_i, t_L(\mathbf{b}_L(\boldsymbol{x}_i) + \boldsymbol{w}\tilde{\boldsymbol{C}})) + \alpha|\boldsymbol{w}|_1 + \beta|\boldsymbol{w}|_2 \tag{7}$$
$$\text{s.t.} \quad \boldsymbol{w}^{\min} \leq \boldsymbol{w} \leq \boldsymbol{w}^{\max}$$

The optimization problem in Equation 7 outputs a single shared set of concept scores that minimizes the proposed loss for a batch of mistakes. Furthermore, we let $\boldsymbol{w}^{\min} = \frac{1}{N}\sum_{i=1}^{N}\boldsymbol{w}_i^{\min}$ and $\boldsymbol{w}^{\max} = \frac{1}{N}\sum_{i=1}^{N}\boldsymbol{w}_i^{\max}$ to make the validity constraints work in the batch-wise setting where $\boldsymbol{w}_i^{\max}$ and $\boldsymbol{w}_i^{\min}$ denotes the validity bounds for the sample $i$.

This formulation results in a more 'holistic' understanding of the model bias compared to the sample-by-sample analysis. In practice, using all of our mistakes in the test dataset, we could run Batch-CCE analysis to provide a global interpretation of the model biases. In Figure 13(c), we provide the performance of Batch Mode CCE over 20 scenarios across different severity levels. The y-axis corresponds to the rank of the target concepts, the x-axis gives the severity of the spurious correlation, the bold line is the median performance across 20 scenarios and the confidence intervals give the lower and upper quartiles. We observe that Batch Mode CCE is able to provide an automated analysis of the model biases using a batch of samples, given that it can identify the spuriously correlated concept as one of the major causes over all model mistakes for the given class.

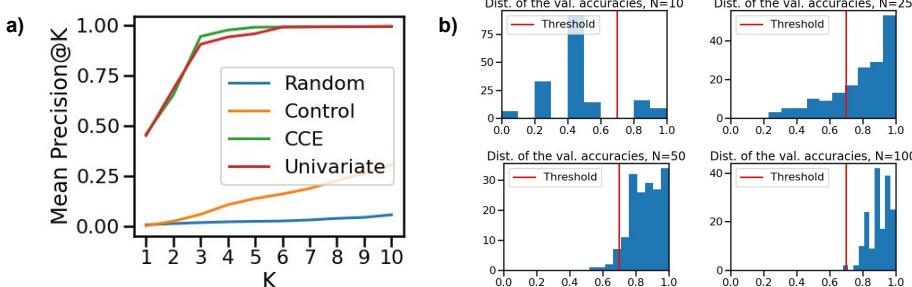

Figure 14: **(a)** We observe how the performance changes with respect to K for the metric Precision@K. **(b)** We provide the distribution of the validation accuracies for each concept, as we vary the number of samples we use to learn the Concept Activation Vectors.

### A.8 DETAILS ON PRECISION@K

In Fig 14(a), we observe how the reported Precision@K metric changes for various values of K. For $K < 3$ the performance is relatively lower. Empirically, we observe that this is mostly due to co-occuring concepts, e.g. for the Dog(Snow) case, sometimes the first two concepts can turn out to be *muzzle* or *dog*. Similarly, for the Dog(Car) case, concepts like *Bus* or *Truck* can turn out to be the Top-2 concepts in the evaluation. We choose $K = 3$ since it is less noisy due to the explained reasons, and also more comprehensible in terms of providing the user a small number of concepts that they can easily work with.

## B CONCEPTS

In Table 5, we list the 170 concepts that we considered, along with their validation accuracies. We kept 168 concepts which had a validation accuracy of 0.7 or greater (**bolded**). We choose the threshold 0.7 since it works well empirically. For a more detailed analysis, in Fig 14(b) we provide the distribution of validation accuracies for concepts, as we vary the number of samples we use to learn the Concept Activation Vectors. As we increase the number of samples, the validation accuracies move beyond the threshold we use (0.7). When we use 100 samples per concept, most of the concepts move beyond this value.

| | | |
|---|---|---|
| loudspeaker(0.86) | microwave(0.82) | arm(0.86) |
| exhaust hood(0.92) | airplane(0.96) | pedestal(0.78) |
| back(0.72) | mouse(0.68) | glass(0.84) |
| polka dots(1.00) | mouth(0.80) | keyboard(0.81) |
| inside arm(0.76) | bird(0.94) | bedclothes(0.82) |
| paper(0.82) | blind(0.86) | brick(0.84) |
| stairs(0.86) | countertop(0.92) | base(0.81) |
| person(0.94) | blueness(0.84) | bathroom s(0.84) |
| pane(0.96) | motorbike(1.00) | hair(0.84) |
| paw(0.84) | candlestick(0.88) | outside arm(0.92) |
| ceiling(0.88) | light(0.88) | street s(0.87) |
| column(0.82) | door frame(0.96) | granite(0.90) |
| cow(0.96) | sand(0.94) | bottle(0.84) |
| cup(0.88) | plate(0.98) | double door(0.92) |
| pillow(0.78) | plant(0.88) | doorframe(0.86) |
| eyebrow(0.88) | flower(0.90) | horse(0.98) |
| toilet(0.90) | ceramic(0.86) | greenness(0.90) |
| back pillow(0.86) | drawer(0.86) | coach(0.92) |
| metal(0.84) | lid(0.90) | bannister(0.86) |
| handle bar(0.78) | fan(0.79) | bush(0.92) |
| blotchy(0.97) | fireplace(0.96) | bowl(0.80) |
| nose(0.80) | leg(0.80) | door(0.82) |
| stripes(0.91) | apron(0.72) | oven(0.80) |
| pack(0.84) | body(0.86) | foot(0.80) |
| frame(0.74) | dining room s(0.88) | board(0.78) |
| bridge(0.82) | sofa(0.78) | bedroom s(0.86) |
| head(0.92) | blurriness(0.95) | footboard(0.80) |
| leather(0.86) | hand(0.90) | fluorescent(0.83) |
| tree(0.98) | knob(0.89) | headlight(0.91) |
| blackness(0.91) | house(0.82) | jar(0.94) |
| mirror(0.94) | pipe(0.82) | bathtub(0.96) |
| flag(0.70) | refrigerator(0.82) | curtain(0.92) |
| book(0.80) | coffee table(0.92) | field(0.90) |
| chandelier(0.94) | cap(0.82) | hill(0.92) |
| ashcan(0.88) | path(0.94) | counter(0.82) |
| cardboard(0.88) | desk(0.84) | balcony(0.88) |
| box(0.72) | napkin(0.79) | ear(0.94) |
| food(0.86) | manhole(0.82) | chest of drawers(0.88) |
| fence(0.82) | building(0.92) | figurine(0.77) |
| lamp(0.96) | basket(0.80) | eye(0.90) |
| ground(0.86) | car(0.90) | cat(1.00) |
| bookcase(0.94) | palm(0.96) | water(0.98) |
| bus(0.96) | laminate(0.82) | handle(0.78) |
| painted(0.88) | dog(0.98) | bumper(0.73) |
| concrete(0.82) | awning(0.66) | clock(0.90) |
| faucet(0.89) | headboard(0.90) | canopy(0.92) |
| bucket(0.74) | drinking glass(0.72) | armchair(0.84) |
| minibike(0.96) | carpet(0.78) | cabinet(0.76) |
| bed(0.77) | earth(0.88) | neck(0.84) |
| can(0.89) | bicycle(0.96) | bag(0.70) |
| chain wheel(0.89) | beak(0.72) | mountain(0.94) |
| redness(0.96) | air conditioner(0.85) | chair(0.86) |
| engine(0.90) | painting(0.80) | grass(0.92) |
| snow(0.84) | pillar(0.84) | chimney(0.73) |
| floor(0.78) | fabric(0.74) | computer(0.86) |
| flowerpot(0.78) | muzzle(0.94) | bench(0.72) |
| ottoman(0.80) | cushion(0.84) | |

Table 5: List of concepts and validation accuracies for SVMs, for a ResNet18 pretrained on ImageNet.

