# OpenReview forum: "Meaningfully Explaining Model Mistakes Using Conceptual Counterfactuals"
_ICLR.cc/2022/Conference — ICLR 2022 Submitted_

### Official Review · Reviewer_KwpD · 2021-10-26

**Correctness:** 4
**Technical Novelty And Significance:** 3
**Empirical Novelty And Significance:** 2
**Recommendation:** 5
**Confidence:** 4

**Main Review:**

Strengths:
-  The problem of explaining a model's mistakes is an important one and the proposed method is clearly defined and well investigated
 with ablation studies.
-  An interesting range of tasks and concepts is considered and, assuming that the group of "OOD images" can be identified ahead of time, the empirical results are convincing.
-  Studying more real/applied datasets such as those in Section 4.4 is an import direction for work on explainability.

Weaknesses:
-  The method from [1] warrants discussion and comparison.  It makes the same assumption as the proposed method (ie, that we have concept labels for each image), provides its own algorithm for finding which concepts a model is relying on, and, further, provides methods for fixing the model when spurious concepts are identified.
-  The experiments show that "for OOD images that are misclassified, the proposed method correctly identifies the concept causing the misclassification."  Unfortunately, this is not particularly useful because defining an "OOD image" requires that we already what spurious concept the model is using, which defeats the point of the proposed method.  Running experiments that start by analyzing the entire set of misclassified images would be more meaningful.
-  While studying explanations in the wild is commendable, these particular experiments are not particularly convincing.
	-  Dermatology:  First, none of these concepts are clinically relevant and, as a result, all explanations using them will indicate some sort of "problem".  Second, the explanations shown in Figure 4 are not particularly convincing because "ashcan" and "water" are frequently relevant concepts; it seems more likely that the explanation method/concept selection are incorrect than the model is genuinely learning to use the presence of "ashcans" to make predictions.
	-  Cardiology:  This has a similar problem to the earlier experiment because most of the concepts indicate some sort of "problem" (only Cardiomegaly and Atelectasis do not).  More generally though, it seems unnecessary to use explainability methods to determine that a model trained on X-Rays taken from one angle won't work as well on X-rays taken from another angle.
	-  Including more clinically relevant concepts or exploring datasets where the Broden concepts are relevant would address this issue.

[1] Singh, Krishna Kumar, et al. "Don't Judge an Object by Its Context: Learning to Overcome Contextual Bias." Proceedings of the IEEE/CVF Conference on Computer Vision and Pattern Recognition. 2020.

**Summary Of The Paper:**

The paper proposes a Conceptual Counterfactual Explanation for explaining a model's mistakes.  It starts by learning a "concept" using a linear separator in the models representation space and a small amount of data annotated with that concept.  Then, it explains a model's mistake by finding a change in those concepts that changes the model's prediction, does not change too many concepts, and does not change the concepts in an unrealistic way.  The method is verified by showing that, for OOD test points where the model makes mistakes, this explanation can identify the concept responsible for those mistakes.

**Summary Of The Review:**

While the proposed method is interesting, the supporting experiments need fundamental changes in order to be impactful.  Specifically, the analysis should not start with the assumption that a set of "OOD images" has been identified and the experiments should include more concepts that are relevant to the predictive task.

# Update after discussion with the authors

I'd like to thank the authors for a very productive discussion and for taking the time to run additional experiments to address my concerns.  At this point, I think that the setup, results, and analysis for the settings where the model is trained on a deliberately biased distribution are convincing.  As a result, I've increased my score to a 5.

However, I am still concerned that using CCE in a less controlled setting will be difficult due to "false positives" for relevant concepts:
-  These appear in Figure 4 in the form of "water" and "ashcan".  In this setting, it's easy to label these as false positives because they are unrelated to the task;  but this seems like more of a reason to exclude those concepts than an actual fix to the problem.
-  As an example of where it is more difficult to determine if something is a "false positive" consider Figure 6.  While it seems likely that concepts like "building" and "flag" are false positives, its much less clear  if "cow" is.  If we consider adding a literal "cow" to the image, it seems unlikely that that would help the model detect the zebra.  But with a more subjective interpretation of emphasizing the "cow shape" of the zebra, then it seems more likely.  Either way, this seems like something that should be tested directly.

One way to address this would be to:
-  Train a model on the entirety of the MetaDataset
-   Find misclassified points and run the batch CCE on them
-   For each of the top concepts, translate the explanation into a hypothesis about about the model's performance on real images (eg, snow is relevant for dog, so the model will perform better on images of dogs with snow)
-  Measure how often those hypotheses are correct

---

> ### Author Response · Authors · 2021-11-16
> **Author response to reviewer KwpD**
>
> Dear Reviewer KwpD,
>
> Thank you very much for your time, review, and feedback. It is very encouraging for us to see that you find the problem important and the ideas interesting. We appreciate the points you raised in your review, we believe these led to important improvements in our paper. Below we provide our response to your points:
>
> **1- Comparison to Singh, Krishna Kumar, et al.:** The most important distinction is that essentially these two methods tackle very different scenarios. Firstly, [1] needs access to the training dataset to quantify bias (Eq1 of the paper). Secondly, it also assumes that concept labels exist for the training data. In practice, it is restrictive to assume access to the training dataset/training procedure, let alone the concept labels for the training points. We often deploy models that are pre-trained on some other task (e.g. transfer learning), or trained with a dataset we do not have access to (e.g. medical applications). In our framework, we do not assume access to the training data, we only assume white-box access to the model being deployed - which is much more practical. Then, we can use any external dataset to annotate concepts using our model - this could be any dataset that contains concepts of interest. We aim to identify biases/artifacts causing the model mistakes without using the training data to label human interpretable concepts. It is also important to note that the common goal with [1] is to address bias in the training dataset - thus we updated our related works section reflecting their valuable work.
>
> **2- “Running experiments that start by analyzing the entire set of misclassified images”:** Thank you very much for raising this very important point. In Appendix A.7.3, we extend CCE to conduct an automated analysis for a set of mistakes. Instead of doing a sample-by-sample analysis, we look for a more holistic method for analyzing model biases. Namely, we take a large set of misclassified images and run CCE to output a single explanation for the entire set of images. We propose Batch-Mode CCE and we demonstrate that it implements a mechanism to identify global explanations of the biases learned by the model. Consequently, we can take any batch of samples from the test distribution to characterize the distribution shift, instead of a manual sample-by-sample evaluation.
>
> **3- Lack of expressivity of concepts in clinical experiments:** We would like to stress that what we aim to do is actually debug and understand why the model is failing. In clinical use cases, the failures due to image artifacts or very biased distributions of metadata constitute a huge obstacle before deploying the models. In our real-world experiments, we showcase the ability of our model to identify these failure modes - rather than actually explaining a diagnosis. More importantly, these are all unmodified real-world scenarios where widely-used models actually fail. A more thorough response can be found in our Overall Response - 2.
>
> Again, thank you very much for your review and suggestions! We believe the points you raised resulted in several important improvements in the paper, please also see our Overall Response. We would appreciate it if you could consider increasing your score given our response and the clarifications we added. Please let us know if you have any additional questions.
>
> [1] Singh, Krishna Kumar, et al. "Don't Judge an Object by Its Context: Learning to Overcome Contextual Bias." Proceedings of the IEEE/CVF Conference on Computer Vision and Pattern Recognition. 2020.

---

> > ### Comment · Reviewer_KwpD · 2021-11-16
> > **Reviewer Response**
> >
> > 1.  It seems possible to calculate Equation 1 from [1] using the setup from this work:
> > -  Instead of using the training data, use the testing data (in the same way the CCE does)
> > -  Instead assuming that the true concept labels are given, use the same predicted concept label (in the same way that CCE does for the "validity" constraint)
> >
> > Perhaps this is too much of an extension of the methodology from [1] to warrant a comparison; I'm not certain that it is, but will leave that up to the other reviewers and the AC.
> >
> >
> > 2. While this is a promising result, it doesn't quite get at my original point for two reasons:
> > -  First, it is still starting from a set of "OOD images" (unless this was changed as well for lower "severities"?)
> > -  Second, it is still analyzing models that have been deliberately trained to make a single (fairly obvious) mistake
> >
> >
> > 3.  Regardless of whether or not the goal is to "understand failures" or "explain diagnoses" (quotes paraphrased), the fact that "ashcans" and "water" are consistently relevant indicates that the method has a high false-positive rate for identifying concepts that are very likely not related to the model's prediction (in this case, because they don't appear in the training data).

---

> > > ### Author Response · Authors · 2021-11-17
> > > **Author response to Reviewer KwpD**
> > >
> > > Response to Reviewer KwpD:
> > >
> > > ​​1- **Suggested bias metric** : We are happy to follow your suggestion. Let $ \textrm{bias}(c, z) = \frac{\frac{1}{|I_{c} \cap I_{z}|} \sum_{i \in I_{c} \cap I_{z}} {\hat{p}(i, c)} } {\frac{1}{|I_{c} \setminus I_{z}|}  \sum_{i \in I_{c} \setminus I_{z}} {\hat{p}(i, c)}}$ (following Equation 1), where z is the category and c is the concept. Assume we are considering the case of dog(snow). This analysis would suggest looking at $\textrm{bias}(\text{snow}, \text{dog})$, and perhaps all other concepts and categories for comparison. We compute these statistics over the test distributions. Here are some numbers:
> > > $\text{bias}(snow, dog) = 0$, $\text{bias}(snow, cat) = 0$, $\text{bias}(snow, bird) = 0$, $\text{bias}(tree, dog) = 0.024$, $\text{bias}(car, dog) = 0.024$, $\text{bias}(sofa, dog) = 0$, $\text{bias}(car, bird) = 0$.
> > >
> > > As demonstrated here, Equation 1 needs access to the training datasets for these metrics to be impactful. Namely, having $\text{bias}(snow, cat)=0$ in the test distribution does not imply anything about the model bias. Only if we had access to the training data, bias(snow, dog) would result in a large number, then by checking the numbers for the test distribution we would understand the shift. Over our concept bank, there might be many concepts that do not show up with a particular category in a test distribution. Thanks for this clarification, and we are happy to further look into this if you have any more suggestions.
> > >
> > > 2.1 - There may be confusion here about how we test our method, please let us clarify how we construct our test distribution: We sample images randomly from the *entire* MetaDataset(Visual Genome), holding the images we used for training the model out. We do not specifically look for “OOD images”. Then, among the mistakes made by the model, we randomly sample 50 of them and examine them. Does this clarify your point?
> > >
> > > 2.2 - It is true that in our controlled experiments, one category is correlated with a single particular concept. The purpose of these experiments is to provide an environment where we can control the correlation and provide concrete results. We can also run this analysis for more than a single concept, i.e. let us generalize this to 2 concepts, for example. E.g. We train the model on dog images, where 50% of images have the concept snow and the other 50% have the concept car. In these scenarios, CCE finds out both of these biases in the model with high precision. Below we report results over a few scenarios:
> > >
> > > $$ \begin{array} {|l|l|}
> > > \hline
> > > \textbf{Scenario}    & \textbf{Precision@5 for Concept 1} & \textbf{Precision@5 for Concept 2} \\\\ \hline
> > > \text{Dog(Snow + Car)} & 1.0   & 0.98         \\\\ \hline
> > > \text{Cat(Bed + Cabinet)}                & 0.98 & 0.98   \\\\ \hline
> > > \text{Cat(Computer + Book)}                & 0.96 & 0.92   \\\\ \hline
> > > \end{array}
> > > $$
> > > Where we have 50 test images for each scenario. It is also possible to generalize this to more scenarios and concepts, however, we believe that our controlled experiments convey the general message that CCE can identify the biases learned by the model.
> > >
> > > 3- We agree that there can be concepts that CCE suggests which do not seem directly relevant to the task. We should keep in mind that there are also similarities among the learned concept vectors. Water or ashcan concepts may be very similar to certain textures that are relevant to the dermatology task, where it was not possible to obtain the true concept itself. This is a general property of concept-based methods.
> > >
> > > However, there are 2 ways to mitigate this:
> > >
> > > 3.1 - Explanations are for humans. For an expert using this model, it is easy to discard the “ashcan” suggestion, where more related “skin color”, “blotchiness” or “defocus blur” concepts are also among the suggestions (mostly with a larger score).
> > >
> > > 3.2 - As the user keeps using concept-based methods, they can understand what type of concepts is needed or relevant to the task. Accordingly, it is possible to update the bank and obtain a more informative bank. Truly, having a rich concept bank is important in this methodology.
> > >
> > > Overall, we agree that there is room for improving CCE. We also want to highlight CCE is the first method of its type and it can open up many new directions of research.
> > >
> > > Thank you for your thorough analysis and time!

---

> > > > ### Comment · Reviewer_KwpD · 2021-11-17
> > > > **Thank you for these clarifications.  They address some of my concerns.  I have a few follow up questions.**
> > > >
> > > >
> > > > 1.  Thanks for looking into this!  At least to me, it seems very unlikely that bias(snow, dog)=0 because this would require that the model's predicted confidence for the class "dog" is (essentially) 0 for every image of a "dog" that also contains "snow".
> > > > -  This might be because your definition is using $\hat{p}(i,c)$ instead of $\hat{p}(i,z)$?  It appears that your definition swapped the position of the "class" and the "concept".
> > > >
> > > >
> > > > 2.1.  Thanks for clarifying!  This alleviates my concern over this.
> > > >
> > > >
> > > > 2.2.  Thanks for running this!
> > > >
> > > > 2.2.1.  My initial reaction to this was that CCE is simply identifying correlations in the training set which, at first glance, doesn't seem that hard.  Until I recalled that CCE doesn't see the training distribution.  This might be worth emphasizing somewhere.
> > > > -  I may have missed it, but do you have a result showing that the model makes more mistakes on images of dogs without snow (or, more generally, on the "OOD images")?  (This probably comes from the results of from our discussion 2.1)
> > > > -  There is some work showing that a correlation is not necessarily sufficient for a model to learn to use a specific "concept" and so having some sort of verification that the model is using the "concept" would be nice.
> > > >
> > > > 2.2.2.  How is "Precision@K" defined for this (and other) experiments?
> > > > -  The definition I'm familiar with is "the fraction of relevant results in the top K" which doesn't seem right in this setting given that there are at most 2 relevant results (Concept1 and Concept 2) and so "Precision@5" should be at most 0.4
> > > > -  Do you define it as the "probability that the relevant concept is within the top K"?
> > > >
> > > > 3.  I agree and think that this probably warrants mentioning somewhere (probably in the appendix).
> > > >
> > > > 3.1.  While domain knowledge can rule these out in this setting, it doesn't seem possible in general whenever the concepts are, at least partially, related to the prediction.  For example, these false-positives may be more problematic if you trained on the entire MetaDataset.

---

> > > > > ### Author Response · Authors · 2021-11-18
> > > > > **Author response to Reviewer KwpD**
> > > > >
> > > > > 1- **Further understanding of the suggested bias metric**: We were using $\hat{p}(i, c)$ and not $\hat{p}(i, z)$, sorry about the confusion. Below we will report some numbers for $\hat{p}(i, z)$.
> > > > >
> > > > > For a model trained on the Dog(Snow) scenario:
> > > > > $\text{bias}(snow, dog) = 2.04$, $\text{bias}(tree, dog) = 1.42$, $\text{bias}(car, dog) = 2.56$, $\text{bias}(mountain, dog) = 2.01$, $\text{bias}(sofa, cat) = 1.32$, $\text{bias}(bed, cat) = 2.24$.
> > > > >
> > > > > It is unclear to us how should one interpret these numbers. Extending this into a larger number of test samples can possibly make this analysis more successful. Namely in our testing procedure, we randomly sample 50 images of dogs from the entire MetaDataset(Visual Genome), and the probability of many of them having a particular concept is not significantly large. For instance, in our case, only 1 of them turns out to have the concept car and 49 of them do not have the concept car, this may be the reason why $\text{bias}(car, dog) > \text{bias}(snow, dog)$. In general, we believe that making this metric work in any setting is not straightforward, albeit not impossible. For instance, we need larger sample sizes and a diverse test distribution to obtain good estimates.
> > > > >
> > > > > CCE is agnostic to the number of samples(can explain a single sample or a batch of samples), does not need to individually go through each (concept, label) subset in the test dataset to make an analysis, and is much more flexible in the sense that it is not just “bias detection” specific, but you can do any form of counterfactual analysis (e.g. CCE can detect that Blur should be removed for a given sample). Given *any* mistake, CCE aims to explain the mistake just using white-box access to the model, without additional information. We believe both approaches are interesting to pursue, however, there are fundamental differences in the way methods approach the problem.
> > > > >
> > > > > 2- **CCE does not see the training distribution**: We also think the fact that CCE does not see the training distribution is crucial, we tried to emphasize this in the Introduction section in the revised version (see *Only needs the model: CCE only needs white-box access to the model. CCE \emph{does not require access to training data}. ..*). We will look into emphasizing this point more as it is one of our key motivations, thank you for this point.
> > > > >
> > > > > 2.1 - **Verification that the model is using the correlated concept**: We believe this is a crucial point to demonstrate, thank you for pointing this out. Here are some numbers, where we have 50 pairs of images for the category of interest with and without the concept and we report the mean accuracy in different scenarios:
> > > > >
> > > > > $$ \begin{array} {|l|l|}
> > > > > \hline
> > > > > \textbf{Scenario}    & \textbf{Accuracy for images with the Concept} & \textbf{Accuracy for Images Without the Concept} \\\\ \hline
> > > > > \text{Dog(Bed)} & 0.86  & 0.46 \\\\ \hline
> > > > > \text{Dog(Snow)} & 0.76  & 0.12 \\\\ \hline
> > > > > \text{Cat(Bed)} & 0.84  & 0.54 \\\\ \hline
> > > > > \text{Cat(Computer)} & 1.0  & 0.68 \\\\ \hline
> > > > > \text{Cat(Keyboard)} & 0.96  & 0.46 \\\\ \hline
> > > > > \text{Dog(Water)} & 0.82  & 0.4 \\\\ \hline
> > > > > \text{Dog(Chair)} & 0.74  & 0.54 \\\\ \hline
> > > > > \text{Cat(Cabinet)} & 0.8  & 0.62 \\\\ \hline
> > > > > \text{Dog(Car)} & 0.86  & 0.66 \\\\ \hline
> > > > > \text{Cat(Grass)} & 0.72  & 0.46 \\\\ \hline
> > > > > \text{Dog(Grass)} & 0.96  & 0.56 \\\\ \hline
> > > > > \text{Cat(Book)} & 0.9  & 0.64 \\\\ \hline
> > > > > \end{array}
> > > > > $$
> > > > > E.g. For the model trained in the Dog(Bed) scenario, the model obtains a 86%(43/50) accuracy over 50 dog images with the bed concept, and it obtains 46%(23/50) accuracy over the dog images without the bed concept. We mostly see a stark difference in the accuracies for images with and without the spurious concept, which provides evidence that models actually rely on the correlated concept.
> > > > >
> > > > > 2.2.2- **How is "Precision@K" defined**: Yes, your latter definition is the metric we use, i.e. “Probability that the relevant concept is within the Top-K”. For instance, in Figure 2, we report “Fraction of samples concept is in Top3” (Precision@3). In the paper, we will try to make this point more clear, thank you!
> > > > >
> > > > > 3- **Discussion on Similarities among concepts**: We agree that it is valuable to include this discussion, thank you for mentioning this. We will also add this point to the paper.
> > > > >
> > > > > 3.1- **Discarding individual suggestions**: We agree with this viewpoint, we should be mindful about finding strategies in assessing false positives. In general, for any interpretability algorithm, there are various things that could be done here both in the human/machine interface and the methodologies themselves, which are further directions CCE could be improved on.
> > > > >
> > > > > Overall, we believe the discussion has been very fruitful. Thank you for helping us improve the paper. We truly appreciate the time you invest in the discussion.

---

> > > > > > ### Comment · Reviewer_KwpD · 2021-11-18
> > > > > > **Many of my concerns have been addressed, I look forward to reading the updated draft and re-assessing my score**
> > > > > >
> > > > > > These reasons all look good/sound like reasonable explanations/make sense, thank you for running these additional tests!  Once the draft has been finalized, I'll re-read it and re-assess my score.

---

> > > > > > > ### Author Response · Authors · 2021-11-20
> > > > > > > **Thank you for your time and comments!**
> > > > > > >
> > > > > > > Dear Reviewer KwpD,
> > > > > > >
> > > > > > > We are very happy to hear your kind comments, thank you very much for your critical suggestions.
> > > > > > >
> > > > > > > We further updated our manuscript following discussions in this thread, in addition to our updates reported before in our Overall Response. Some of them are listed below for convenience:
> > > > > > >
> > > > > > > 1) In Appendix A.7.3, you can find our batch evaluations of a set of mistakes.
> > > > > > >
> > > > > > > 2) In Appendix A.4.1, you can find our discussion on explanation artifacts and concept similarities.
> > > > > > >
> > > > > > > 3) In Appendix A.2, Table-3, you can find the accuracies evaluated over images with and without the particular concept, for verification that the model is using the concept.
> > > > > > >
> > > > > > > 4) In the writing, we aimed to better emphasize a) The details of the Precision@3 metric b) the fact that we are not using the training dataset c) Our evaluation protocol.
> > > > > > >
> > > > > > > Once again we really appreciate the valuable time you invested in the rebuttal period. Please let us know if there are any further points, we are looking forward to your assessment.

---

> ### Author Response · Authors · 2021-11-23
> **Author response to reviewer update**
>
> Dear Reviewer KwpD,
>
> Thank you for reviewing the updated paper and reassessing your score. Following the point you make in your latest comment, we have done the suggested experiment.
>
> Particularly:
>
> 1- We trained a model on the entire MetaDataset, without introducing a correlation.
>
> 2- We find 50 misclassified images for four classes and run Batch CCE on these images.
>
> 3- We report Top-5 concepts for each of the classes:
>
>
> $$ \begin{array} {|l|l|}
> \hline
> \textbf{Class} & \textbf{Top5 Concepts} \\\\
> 			\hline
> 			\textbf{Dog} & \text{Dog, Muzzle, Motorbike, Bicycle, Bed Clothes }\\\\
> 			\hline
> 			\textbf{Cat} &  \text{Cat, Inside Arm, Door Frame, Cabinet, Chest of Drawers}\\\\
> 			\hline
> 			\textbf{Bird} &  \text{Airplane, Bird, Water, Flag, Balcony}\\\\
> 			\hline
> 			\textbf{Elephant} & \text{Horse, Cow, Blotchy, Canopy, Concrete}\\\\
> 			\hline
> 	\end{array}
> $$
>
>
> Here, due to time restrictions until the end of the rebuttal period, we cannot provide a quantitative analysis of how this model would perform on each of these subsets. However, qualitatively, we observe that these are the concepts that categories may easily be correlated with. I.e. it is easy to imagine that *inside arm* and cat, or *muzzle* and dog, or *airplane* and bird appear frequently together in the same image / are well-represented combinations in the training distribution. We believe this experiment provides another piece of evidence that shows CCE performs reasonably in less-controlled settings as well. Thank you for this suggestion.
>
> Please let us know if these address your concern, we are happy to follow up. Thank you again for your time.

---

### Official Review · Reviewer_GEvb · 2021-11-01

**Correctness:** 4
**Technical Novelty And Significance:** 3
**Empirical Novelty And Significance:** 3
**Recommendation:** 6
**Confidence:** 4

**Main Review:**

Pros:
 - Well written and clearly structured
 - Interesting and promising approach to compute counterfactuals in the domain of concepts

Cons:
 - Evaluation was done manually - i.e. manual inspection of samples. I understand that an automatic evaluation is challenging and I also appreciate that the authors are aware if this and suggest an user study - however, still I think that this manual inspection as an evaluation is a major limitation of this work. I would be more convinced if the authors would have done the evaluation as a proper user study.
 - While the examples in the paper look convincing, I was wondering whether it is possible to come up with formal guarantees - i.e. are there any formal guarantees that the final explanation is valid? The authors added some validity constraints but to me they look like a heuristic without any formal guarantees - I encourage the authors to comment on this.

**Summary Of The Paper:**

The authors propose a method for computing concept based counterfactual explanations - i.e. using the presence or absence of concepts (human understandable) for explaining the models prediction. In the paper they focus on image classifiers.

**Summary Of The Review:**

Overall an okay paper with some very interesting ideas, although I have some concerns regarding the evaluation of the method (see main review).

---

> ### Author Response · Authors · 2021-11-16
> **Author response to reviewer GEvb**
>
> Dear Reviewer GEvb,
>
> Thank you very much for your time, review, and comments. We are very encouraged to hear that you find the paper “well written and clearly structured”, and the ideas “interesting and promising”. Below we provide our responses to the points you raised:
>
> **1- Manual evaluation:** Thank you very much for raising this important point. In Appendix A.7.3, we extend CCE to conduct an automated analysis for a set of mistakes. Instead of doing a sample-by-sample analysis, we look for a more holistic method for analyzing model biases. Namely, we take a large set of misclassified images and run CCE to output a single explanation for the entire set of images. We propose Batch-Mode CCE and we demonstrate that it implements a mechanism to identify global explanations of the biases learned by the model. Consequently, we can take any batch of samples from the test distribution to characterize the distribution shift, instead of a manual sample-by-sample evaluation.
>
> **2- Formal guarantees:** We believe that providing formal guarantees to ensure that the perturbations in the concept space are not violating real-world conditions is a very important question. We quantify the extremums in the real world conditions using statistics of margins for the concept samples, which empirically performs very well. However, we believe a more formal analysis of validity would on its own be a very important direction for future work in concept-based explanations methods in general.
>
> Again, thank you very much for your review and suggestions! We are pleased to improve our paper following your valuable feedback. We would appreciate it if you could consider increasing your score given our response and the new experiments we added, please also see our Overall Response. Please let us know if you have any additional questions

---

> > ### Comment · Reviewer_GEvb · 2021-11-16
> > **Thank you for your clarifications.**
> >
> > Thank you for your clarifications.

---

### Official Review · Reviewer_nQZu · 2021-11-02

**Correctness:** 3
**Technical Novelty And Significance:** 4
**Empirical Novelty And Significance:** 2
**Recommendation:** 6
**Confidence:** 4

**Main Review:**

Strengths:

- Novel combination of concept activations and counterfactual explanations
Excellent writeup of approach and motivation

- Interesting controlled evaluation of approach that demonstrates effectiveness.

Weaknesses:

- Approach cannot be generalized to obtain explanations for a true test image since approach requires label y to obtain perturbation. How is it possible to make the approach work without the label y?

- Lack of baselines in controlled evaluation (sections 4.1 to 4.3): consider the following baseline, take the test example, find the closest correctly classified example to it in a validation set (that has same distribution as training data), and subtract the concept vectors for the test image - closest image in validation. Then show the top 3 concepts of the difference.

- Lack of ability to visualize perturbed test examples. How is it possible to visually see what the test image with the perturbation looks like? Perhaps this is easy, but would appreciate it if the authors can elaborate on how to do this.

- Lack of discussion surrounding the requirement of being able to collect images in the domain of interest that describe each of the concepts independently of the trained data. For example, in the dog and snow example, to detect the concept of snow, the SVM must have seen images of snow with and without dogs.

- In controlled evaluation, the cause of the mistake can always be found in the concepts. What happens if the mistake cannot be attributed to a concept? Does the method say that none of the concepts explain the mistake or does it falsely attribute it to something else? One can do this evaluation by removing e.g. “snow” from the concept bank and running the same experiment.

- Lack of objective evaluation metric and baseline in the “in the wild” evaluation: I understand the lack of baselines in the controlled study, but for the evaluation with clinicians, it would have been helpful to see if with other baselines (maybe something like GradCAM?) the clinicians can try to identify on their own the reasons they think the model got it incorrectly and see if they match with the concepts retrieved.

- The clinical concepts especially for the chest x-ray use case are not comprehensive enough to be sufficiently descriptive. I don’t believe in it’s current form, the in the wild evaluation adds value to the paper.


After author response:

- I have increased my score as the authors have raised some of my concerns, particularly on when the concept is not in the library

**Summary Of The Paper:**

Goal: provide a human readable explanation to why a given classifier made a mistake on a given example.

Approach:

Step1: For the given input domain (here images), first obtain a set of concepts that are human interpretable. These concepts each describe a particular aspect of an image, and a human is expected to be able to look at image and say whether a particular concept applies or does not apply to an image (on scale of [0,1])

Step2: learn a map from the input domain to the concept space using an SVM

Step3: generate counterfactual perturbation of example that ensures that :1) classifier is now correct on example, 2) perturbation is valid (i.e. realistic) and 3) perturbation changes few concepts. The perturbation is the addition of a linear combination of concepts by different amounts.  Key to this is that generating the perturbation requires the true label and query access to the classifier.

Step4: present the user the weight vector of the concept perturbations.

Validation: two ways to evaluate, one controlled and one in the wild.

Controlled evaluation: using dataset of animal images in different settings, confound the class label, e.g. dog, with background, e.g. snow, then ask if approach can recover the confounding factor, here snow. They show that on a median basis, the method finds the right confounding factor in the top 2 spots.

In the wild evaluation: on dermatology and chest x-ray datasets, they studied the mistakes of of a  trained model. The evaluation of their findings is based on discussion with clinicians.


**Summary Of The Review:**

The paper provides a novel way to explain a classifier's mistakes on a test example when given it’s label in terms of a known set of concepts. The paper is very well written and the controlled evaluation demonstrates that it is effective when the concepts can describe the mistakes perfectly. However, the evaluation lacks baselines and results when some of the assumptions fail (concepts don’t describe mistakes). Furthermore, it is hard to understand how the “in-the-wild” evaluation shows the effectiveness of the method. Finally, the method requires knowing the label of the example and thus is more of a debugging approach rather than an explanation method.

In its current form, I think the paper is borderline accept*, however, if the authors can provide answers to some of the weaknesses I presented, I am more than willing to move the score up to an accept as I think the ideas and the problem are extremely interesting.

---

> ### Author Response · Authors · 2021-11-16
> **Author response to reviewer nQZu - Part 1**
>
> Dear Reviewer nQZu,
>
> Thank you very much for your thorough review! We truly appreciate your assessments of our paper as a “novel approach” and an “excellent writeup”. We are very encouraged to see that you also find the ideas and the problem “extremely interesting”. We believe your comments have been very helpful in crucially improving our paper, and below we provide our responses to the points you raised:
>
> **1- Requiring the label**: We would like to highlight that the main motivating use case of our method is to explain mistakes, and a mistake is not defined without a label. That being said, there are various other forms of analysis that can be conducted using our framework. For instance, we could instead use the model prediction as the pseudo-label, try to maximize the cross-entropy loss. This would result in an adversarial attack in the concept space, and we could use this to understand the model behavior. However, these analyses are beyond the main motivation of our paper, which is to understand and explain systematic mistakes made by a model.
>
>
> **2- Baselines**:
>
> 2.1 - In general, concept-based explanations can do a much better job in communicating mistakes by the model. Saliency-based methods can be tricky to interpret in many cases, and especially insufficient in finding biases. We demonstrate a few examples of this in a new figure(Appendix-A4 Figure 9), where we show how GradCAM++ fails to identify/communicate the bias causing the mistake.
>
> 2.2 - The baseline you mention requires “a validation set (that has same distribution as training data)”. Throughout all our experiments, we assume that we do not have any access to the training distribution - only to the model. Given access to the training distribution, there can also be other ways to quantify the bias. In general, we do not know any other methods evaluating the same setting and using concepts, which limits the ability to run further baselines.
> Even though it assumes a different scenario, we implement the suggested baseline in our MetaDataset experiments. Concretely: We take the test image, find the nearest neighbor in the validation distribution (50 images sampled from the same distribution as training) using model embeddings. Subtract the embeddings for the test image from the nearest neighbor, and get Top-3 concepts with the largest cosine similarity to the difference in the embeddings. A comparison in performance can be found below, which are averaged over 20 scenarios:
>
>
> $$ \begin{array} {|l|l|l|l|l|}
> \hline
> \textbf{Method}    & \textbf{Precision@3} & \textbf{Precision@5} & \textbf{Precision@10} & \textbf{Precision@20} \\\\ \hline
> \text{Suggested Baseline} & 0.02                 & 0.03                 & 0.06                  & 0.12                  \\\\ \hline
> \text{CCE}                & \textbf{0.95}        & \textbf{0.98}        & \textbf{0.99}         & \textbf{0.99}         \\\\ \hline
> \end{array}
> $$
>
>
>
> Ultimately, we observe that it is not trivial to obtain these results, even when we have the training data.

---

> > ### Author Response · Authors · 2021-11-16
> > **Author response to reviewer nQZu - Part 2**
> >
> > **3- Visualizing perturbed samples**: There can be a few ways to address this. One way to understand the perturbed image can be to visualize a few nearest neighbors in the embedding space. Another one could be to use a DeepDream[1] variant to perturb the input image, similar to what is done in the original TCAV paper. We believe visualization of the perturbed samples is something important to establish the human-machine interface. It’s a good direction for future work.
> >
> > **4- Importance of the concept library**: We agree that the quality of the concept library is a crucial aspect of our method, just as any other CAV method. We updated our writing reflecting this notion, and a more detailed response can be found in our Overall Response - 3.
> >
> > **5- When the cause is not in the concept bank**: We believe this is a very important issue, thank you very much for raising this. We added a new set of experiments to Appendix-A.7 that follows the scenario that you described.  In these cases, CCE identifies concepts that are "hinting" to the target concept, even though the target concept itself is missing from the bank (ex. When bed is missing, CCE suggests {bed clothes, sofa, headboard, ..}). This provides another piece of evidence that highlights the importance of the richness of the concept bank. If we have a rich concept bank, then CCE helps us identify the spuriously correlated concept even if the exact concept is not directly in the bank.
> >
> > **6- Lack of expressivity of concepts in clinical experiments**: We believe this is also related to your comment about the method being a debugging method, rather than an explainability method. We agree with this notion; what we aim to do is actually debug and understand why the model is failing. In clinical use cases, the failures due to image artifacts or distributions of metadata constitute a huge obstacle before deploying the models. In our real-world experiments, we showcase the ability of our model to identify these failure modes - rather than trying to explain a diagnosis. A more thorough response can be found in our Overall Response - 2.
> >
> > Again, thank you very much for your review and suggestions! We believe your comments led to crucial improvements in our paper, please also see our Overall Response. We would appreciate it if you could consider increasing your score given our response and the new experiments and clarifications we added. Please let us know if you have any additional questions
> >
> > [1] Mordvintsev, Alexander, Olah, Christopher, and Tyka, Mike. Inceptionism: Going deeper into neural networks. Google Research Blog.

---

> > > ### Comment · Reviewer_nQZu · 2021-11-21
> > > **Response to author comment**
> > >
> > > Thank you for your responses. I really appreciate the additional experiment to answer "When the cause is not in the concept bank" and have raised my score because of your response.
> > >
> > > 1. when using the model prediction as the label, why is this adversarial? Shouldn't the concepts help further exaggerate why the model made this prediction?
> > >
> > > I would really have liked a more thorough answer to this question in my review:
> > >
> > > "Lack of discussion surrounding the requirement of being able to collect images in the domain of interest that describe each of the concepts independently of the trained data. For example, in the dog and snow example, to detect the concept of snow, the SVM must have seen images of snow with and without dogs."
> > >
> > > I believe the missing piece in this paper is a theoretical formalism for the assumptions to be able to describe model errors using the concept library. Particularly also on how one can learn the map from input to concepts given a certain data sample. This will require integrating model learning & concept learning into the formulation.

---

> > > > ### Author Response · Authors · 2021-11-22
> > > > **Thank you for your follow-up!**
> > > >
> > > > Dear Reviewer nQZu,
> > > >
> > > > Thank you for your follow-up, we are happy to see that our new experiments partly address your concerns.
> > > >
> > > > Being adversarial:
> > > > There are 2 major points about this which we state below, but briefly, we believe this method would not work in our desired use case.
> > > >
> > > > 1- In our response, we specifically mentioned the case where we “maximize” the cross-entropy loss. This would mean answering a query about “what should we add to / remove from the input the flip the model prediction”, and this is particularly the question adversarial attacks are trying to answer. What you mention would be the case in which we “minimize” the loss, as you mention this may help to “exaggerate why the model made this prediction”.
> > > >
> > > > 2- We believe this may be an interesting question to investigate, however, our main motivation in this paper is to “debug” the mistakes made by the model. This analysis is not possible by your suggestion. For instance, assuming in the dog(snow) scenario we have the model prediction as “cat”. Then, running what you suggest is likely to add concepts such as “cat”, “bed”, “person”, “toy” etc. and it cannot remove the concept “snow” since it is already prevented by the validity constraint. We would not have a way of understanding why the model is missing the category “dog”; we could only understand what the model needs to predict the “cat” better. Hence, this does not address our main motivation of debugging mistakes & understanding model biases.
> > > >
> > > > Regarding concept learning:
> > > > We agree that to learn a concept reliably, we need diverse positive examples of the concept (e.g. all snow/dog images shouldn't have both snow and dog). This is reasonable in practice since the # of positive examples needed is small, and our several experiments demonstrate this point (throughout our work, we use 25-100 positive images in different experiments). For instance in natural images, there are several sufficiently rich datasets to obtain such examples (Visual Genome, BRODEN, ImageNet, ..). In other domains, it is not impractical/extremely burdensome to label 10s of images. Clinicians are usually more than interested in doing this if the method can bring about an understanding of the mistakes of the model being deployed. Importantly, note that this whole procedure does not need to be executed separately by each clinician. If someone introduces positive samples of a concept / labeled database, then the same bank can be shared across different hospitals/users to implement their own pipelines (e.g. Fitzpatrick for skin color). Finally, we also agree that more theoretical analysis is missing in the wider concept-learning literature, and it is an important separate direction. However, as our main motivation is not providing a new concept learning algorithm, we believe this is not in the scope of this work.
> > > >
> > > > Thank you for your time and response, please let us know if you have any remaining questions, we are very happy to follow up.

---

> ### Author Response · Authors · 2021-11-21
> **Dear nQZu: we'd love to hear if you have any further Qs after our response**
>
> Dear reviewer, thank you for your comments, they have really helped us to improve the paper! We hope our response and updated paper addressed your questions. Please let us know if you have any remaining feedback and we are very happy to follow up.
>
> Thank you again for your time!

---

### Official Review · Reviewer_oufr · 2021-11-03

**Correctness:** 3
**Technical Novelty And Significance:** 3
**Empirical Novelty And Significance:** 2
**Recommendation:** 6
**Confidence:** 4

**Main Review:**

Strengths:

- The paper is clearly written, and the figures are demonstrative and clear.

- In section 4.1, the experiments to identify spurious correlations by CCE a good beginning, but the interventions on the MetaDataset images are large drastic effects.  I would lke to see CCE applied to the identificaiton of more subtle spurious correlations that are more realistic.  Perhaps CelebA could be a good starting point?
  - e.g where more subtle latent factors explain the spurious correlation
  - where label imbalances exacerbate spurious correlation by diminishing the ability to estimate concept vectors

- The granny smith colour ablation test in section 4.2 a nice clear example of how the tool can clearly demonstrate a relationship for one given spurious correlation, but it does not tell me much about how robust it is to reporting spurious correlation CAVs at test time, which is the more important problem.

Weaknesses:

- I feel the authors underplay the complexity and investment to define a concept library, especially for specialize application areas.  In diagnostic imaging, prior taxonomies of concepts exist, but can be very difficult to identify in training images (cf. Oakden-Raynor et al.).  In other fields there may not even yet be generally agreed upon ontologies of concepts.  This drawback of the CAV method should be noted.

- Figure 1c is not a particularly good example of explaining a model mistake.  While the *label* given in the image may be “Zebra”, there is clearly a crocodile visible in the image (left of the smaller, left-most zebra).  This image is indicative of a different problem, which is that many natural images are better described as multi-label, as they contain multiple distinct concepts or entities within them.


- Towards the end of section 2 the authors claim
  > "we extend these perturbations to unstructured data (specifically natrual images), whereas existing counteractural explanation methods have been largely confined ot tabular datasets since the perurbations have consitted of changing values in the original low-level feature space"
  This is far from true.  See Singla et al 2019, Goyal et al. 2019, and many other papers that apply counterfactual explanation to latent spaces of models that generate natural images.  This is a young but growing subfield that should be cited to acknowledge their contributions.

- The end of section 3.1 suggests concepts are validated by a hold out set, where those that do not meat a threshold on accuracy are discarded.  How was this threshold determined?  Is it affected by noisy data in the wild?  Is there some clear threshold in the distribution of accuracies over concepts?  And how is this affected by either noise or imbalance in classes of the data?  There are several such questions here that deserve more attention.

- Minor point, but Algorithm 1 should specify that the terminating condition of the while loop is **while** *w not converged* **do**


- Is the definition of the geometric margin ot the decision boundary in section 3.2 quite right?  It is defined as $d_i = \mathbf{c}_{i}b_{L}(\mathbf{x}_i)^T$.  This seems to use $i$ to index both concepts and datapoints in the training set for concepts.

Questions:

- Section 3 defines the *bottleneck layer* as that layer which will be employed to act as feature vector for learning CAVs.  Does the choice of layer matter here?  Or is there any reason to deviate from the pre-activation features from the ultimate fully connected layer?

- For Table 1: Is there a reason for choosing precision @3?  What does the distribution of the precision @K look like for various K?  Is there a steep drop after 3, or is 3 motivated by the desire to provide a user with a more manageable number of possible concepts?


**Summary Of The Paper:**

Explaining errors using semantically meaningful concepts is a critical tool for improving the performance, robustness, and trustworthiness of machine learning models deployed in application areas everywbere.  While there have been several proposed lines of work in this area, none is yet accepted as standard practice.  The authors present a method combining two prior methods in explainability: counterfactual explanation and concept activation vectors, and meld them together as coneptual counterfactual explanations.

They describe the training procedure, and show some examples on both ImageNet and real diagnostic imaging applications.

**Summary Of The Review:**

While this paper combines two complementary techniques in counterfactual explanation and CAVs, I would like to see CCE compared to other counterfactual explanation methods that are trained on natural images and provide semantic highlights or perturbations.  I also think that several important details have been omitted from the discussion (see Weaknesses section), and that in its present form, the work is not ready for publication.

Update: In response to the authors' extensive revisions to address my concerns (and those of the other reviewers), I have increased my score.

---

> ### Author Response · Authors · 2021-11-16
> **Author response to Reviewer oufr**
>
> Dear Reviewer oufr,
>
> Thank you very much for your time, well-thought review, and feedback. We appreciate your assessment of our paper and constructive feedback that led to very impactful improvements in our paper. We provide our responses to your points below:
>
> Comments:
>
> **1- Importance of the concept library:** We agree that the quality of the concept library is a crucial aspect of our method, just as is the case with any other method that utilizes concept activation vectors. We updated our writing reflecting this notion, and a more detailed response can be found in our Overall Response - 3.
>
> **2- More subtle shifts:** This is an important addition to the paper, thank you for raising this. We added a new set of experiments where we vary the level of severity of the correlations in the training distribution. More details can be found in Overall Response - 1.2.
>
> **3- Example in Figure 1c:** Thank you very much for pointing this out. We updated the mistake example with one that better reflects the problem we aim to tackle.
>
> **4- Missing related works in counterfactual explanations in natural images:** Thank you very much for raising this issue. We understand how the previous version of our wording is misleading and thus updated our related works section reflecting some of the exciting work being done in applying counterfactual explanations to natural images. We also highlight our innovations compared to those works. Briefly: Goyal et al. modify the query image using a distractor image to flip the model prediction.  In our case, in order to identify the biases in the model, this requires finding images from the training distribution (concept distractors) and violates our assumptions that we do not have examples from the training data. Singla et al. perturb the input image to flip the model prediction, which requires analyzing the perturbation in the input space. Our work differentiates from them since we are using high-level concepts to do the counterfactual evaluation, instead of perturbation understanding in the input space, and we do not assume access to any data from the training distribution.
>
> **5- Choice of validation accuracy threshold for concepts:** We appreciate your point regarding how concepts are picked according to the validation accuracies of SVMs. We added Figure 14 to Appendix-B demonstrating the distribution of validation accuracies. We see that using a larger number of images to learn concepts improves the validation accuracies. As we increase the number of samples used, almost all of the concepts move beyond the threshold we used (70%). This also relates to the question regarding the noise, as a smaller sample size would mean larger noise. It is worth mentioning that during experimentation, we tested different thresholds and this threshold produced the best empirical results.
>
> **6- Typos:** Thank you very much for pointing out the typos in the convergence statement and the definition of the geometric margin. These are fixed in the updated manuscript.
>
> **7- Choice of layers:** This is a general consideration of all CAV-based approaches. As reported by the original TCAV paper, lower-level layers contain more predictive information regarding low-level features, and higher layers can be more predictive of high-level features. Similar considerations can be applied according to the domain of application.
>
> **8- Choice of K for the Precision@K metric:** Thank you for raising this point. For clarification, we added another Figure in Appendix-A.8 that demonstrates how Precision@K changes with respect to K. As you correctly point out, our initial motivation for providing a low number (K=3) is “to provide a user with a more manageable number of possible concepts”. Furthermore, it is true that at K<3 the performance is relatively lower. This can be due to co-occurring concepts, e.g. for the Dog(Snow) case, sometimes the first two concepts can turn out to be “muzzle” or “dog”, etc.
>
> Again, thank you very much for your review and suggestions! We believe the changes in the paper are important improvements, please also see our Overall Response. We would appreciate it if you could consider increasing your score given our response and the clarifications we added. Please let us know if you have any additional questions.

---

> ### Author Response · Authors · 2021-11-21
> **Dear oufr: we'd love to hear if you have any further questions after our response**
>
> Dear reviewer
>
> Thank you very much for your helpful feedback and suggestions, they helped us to improve the paper. We tried to carefully address your comments in our response and the updated paper. Please let us know if you have any further questions, and we are very happy to follow up!
>
> Thank you for your time!

---

### Author Response · Authors · 2021-11-16
**Overall Response - Part 1**

Thank you very much to all reviewers for their constructive feedback. Below we give a summary of our major updates to the paper:


1- **New Experiments:** In Appendix A.7, we provide more challenging scenarios for CCE to address several concerns raised by reviewers.

1.1- **When the cause is not in the concept bank:** In Appendix A.7.1, we create scenarios where the target concept does not exist in the concept bank, as suggested by Reviewer nQZu. In these cases, CCE identifies concepts that are "hinting" to the target concept, even though the target concept itself is missing from the bank(ex. When bed is missing, CCE suggests {bed clothes, sofa, headboard, ..}). This provides another piece of evidence that highlights the importance of the richness of the concept bank. If we have a rich concept bank, then CCE helps us identify the spuriously correlated concept even if the exact concept is not directly in the bank.

1.2 - **Less drastic and more realistic spurious correlations:** In Appendix A.7.2, we move to less extreme spurious correlations. Namely, in our controlled experiments, we vary the severity of the correlation in the training dataset (fraction of images that have a particular concept in the training distribution, e.g. {10, 20, 50,..}% of dog images having the snow concept). As raised by Reviewer oufr, this is a less drastic and more challenging case that is closer to real life. We show that CCE performs well even in these smaller spurious correlations.

1.3 - **Automated analysis of mistakes with Batch-Mode CCE:** In Appendix A.7.3, we extend CCE to conduct an automated analysis for a set of mistakes. As pointed by multiple reviewers, instead of doing a sample-by-sample analysis, we look for a more holistic method for analyzing model biases. Hence, we propose Batch-Mode CCE and we demonstrate that it implements a mechanism to identify global explanations of the biases learned by the model. Consequently, we can take any batch of samples from the test distribution to characterize the distribution shift, instead of a manual sample-by-sample evaluation.

2- We believe it is very important to emphasize that what we are doing is not directly explaining why a model made a particular decision. In this sense, we agree with the comments arguing that our method “is more of a debugging approach rather than an explanation method”, and “all explanations using them{concepts in the clinical experiments} will indicate some sort of ‘problem’”. We are trying to debug the model to understand why a model made a particular mistake. We aim to identify artifacts in the datasets or models that are frequently causing errors in the real world. In clinical images, it is very common to have perturbations as artifacts, and this is an important problem to address [1]. In other settings, the gender of the patient [2] or the view of the Chest X-Ray [3] can cause problems, let alone clinically challenging samples. Although our framework can be used to understand the mistakes in the diagnosis that is due to medical concepts, the idea of our experiments is to show that CCE can be used to identify issues in the model training and evaluation pipelines. Our experiments are unmodified real-world cases, where CCE helps uncover these issues. Without CCE, it would require individually analyzing the skin-color distribution per each skin condition in the training distribution to find the bias, and it is impractical to assume access or exhaustively try to quantify the bias in each (meta-label, class) pair. Additionally, following this discussion, we change our title to **Meaningfully Debugging Model Mistakes Using Conceptual Counterfactual Explanations**.


3- As pointed out by multiple reviewers, the collection of the concept bank is a crucial aspect of the methodology. We updated our paper to highlight the importance of this aspect. However, it is important to note that this is not a severe limitation. CAVs have been deployed in various settings from dermatology[4], EHRs[5], histopathology[6], and many more. Firstly, it is worth mentioning that this is a general property of the CAV methodology, rather than being unique to our explanation algorithm. Secondly, in our in-the-wild experiments, we demonstrate that using a sample size as low as 25, we can still obtain good results. While acknowledging this important consideration, we would like to emphasize that in many practical scenarios this limitation is feasible to address.

---

> ### Author Response · Authors · 2021-11-16
> **Overall Response - Part 2**
>
> 4- We updated our related works section reflecting comparisons to existing works in counterfactual explanations for natural images and overcoming contextual bias papers. Our work is unique in several ways compared to these approaches: We do not assume access to the training distribution, any samples from the training dataset, or the training procedure, but we only assume white-box access to the trained model.
>
> 5- We provide justifications for our choices of validation accuracy thresholds and the sensitivity of Precision@K metric to the selection of K in Appendix B and Appendix A.8 respectively.
>
> 6- In a new figure (Appendix A.4 Figure 9) we compare a few examples of class activations maps based explanations in the dermatology setting, and clarify why these methods cannot be trivially used to identify and communicate biases in the model.
>
> Overall, we are very encouraged to see the stimulating discussion around the paper and happy to have added the new experiments, which we believe made very crucial improvements to our work. Once again, we would like to thank all reviewers for their thorough analysis and comments. We would really appreciate it if you could let us know whether our response clarifies the points you raised.
>
> [1] Muraco L. Improved Medical Photography: Key Tips for Creating Images of Lasting Value. JAMA Dermatol. 2020;156(2).
>
> [2] Larrazabal et al. Gender imbalance in medical imaging datasets produces biased classifiers for computer-aided diagnosis. Proceedings of the National Academy of Sciences Jun 2020.
>
> [3] Wu, E., Wu, K., Daneshjou, R. et al. How medical AI devices are evaluated: limitations and recommendations from an analysis of FDA approvals. Nat Med 27, 582–584 (2021).
>
> [4] Lucieri, Adriano et al. “On Interpretability of Deep Learning based Skin Lesion Classifiers using Concept Activation Vectors.” International Joint Conference on Neural Networks (IJCNN) (2020).
>
> [5] Mincu et al. “Concept-based model explanations for electronic health records.” In Proceedings of the Conference on Health, Inference, and Learning (CHIL '21).
>
> [6] Graziani et al. “Regression Concept Vectors for Bidirectional Explanations in Histopathology” MLCN/DLF/iMIMIC@MICCAI 2018.

---

### Decision · Program_Chairs · 2022-01-20

**Decision:**

Reject

**Comment:**

This paper proposes a novel method called CCE for explaining mistakes by DNNs on image classification. It is built on top of two prior ideas: counterfactual explanations and concept activation vectors. CCE explains a mistake by assigning scores to a shot list of concepts, where a large positive score means that adding that concept to the image will increase the probability of correctly classifying the image, as will removing or reducing a concept with a large negative score.
The strengths of the paper include novel combination of previous work, clear presentation, interesting experiments, convincing results on controlled settings.  The weaknesses include the lack of results on less controlled settings, the lack of more meaningful spurious correlations in the medical examples, and the lack of user studies.
Although the reviewers have shown interests in this paper, they clearly do not support the paper strongly. In addition, the authors have missed the following paper that also combines counterfactual explanations and concept activation vectors:
Akula, Arjun, Shuai Wang, and Song-Chun Zhu. “Cocox: Generating conceptual and counterfactual explanations via fault-lines.” Proceedings of the AAAI Conference on Artificial Intelligence. Vol. 34. No. 03. 2020.